# GeneMAN: Generalizable Single-Image 3D Human Reconstruction from Multi-Source Human Data

**Wentao Wang[1]\***, **Hang Ye[2]\***, **Fangzhou Hong[3]**, **Xue Yang[4]**, **Jianfu Zhang[4]**,
**Yizhou Wang[2]**, **Ziwei Liu[3]**, **Liang Pan[1]†**

[1]Shanghai AI Laboratory    [2]Peking University
[3]Nanyang Technological University [4]SAIS & SCS, Shanghai Jiao Tong University

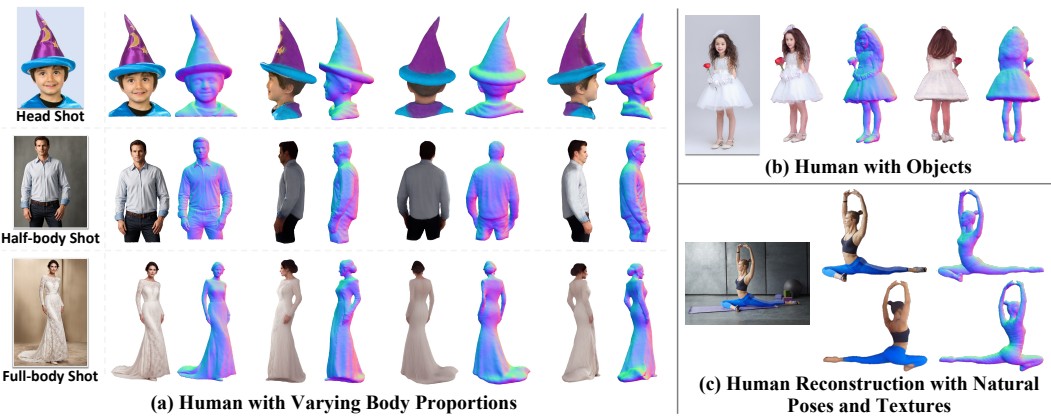

Figure 1: **GeneMAN is a generalizable framework for single-view-to-3D human reconstruction, built on a collection of multi-source human data.** Given a single in-the-wild image of a person, GeneMAN could reconstruct a high-quality 3D human model, regardless of its clothing, pose, or body proportions (*e.g.*, a full-body, a half-body, or a close-up shot) in the given image. The anonymous project page of GeneMAN is: https://roooooz.github.io/GeneMAN/.

## Abstract

Given a single in-the-wild human photo, it remains a challenging task to reconstruct a high-fidelity 3D human model. Existing methods face difficulties including a) the varying body proportions captured by in-the-wild human images; b) diverse personal belongings within the shot; and c) ambiguities in human postures and inconsistency in human textures. In addition, the scarcity of high-quality human data intensifies the challenge. To address these problems, we propose a **Gene**ralizable image-to-3D hu**MAN** reconstruction framework, dubbed **GeneMAN**, building upon a comprehensive multi-source collection of high-quality human data, including 3D scans, multi-view videos, single photos, and our generated synthetic human data. GeneMAN encompasses three key modules. **1)** Without relying on parametric human models (*e.g.*, SMPL), GeneMAN first trains a human-specific text-to-image diffusion model and a view-conditioned diffusion model, serving as GeneMAN 2D human prior and 3D human prior for reconstruction, respectively. **2)** With the help of the pretrained human prior models, the Geometry Initialization-&-Sculpting pipeline is leveraged to recover high-quality 3D human geometry given a single image. **3)** To achieve high-fidelity 3D human textures, GeneMAN employs the

---

\*Equal contributions.
†Corresponding author.

39th Conference on Neural Information Processing Systems (NeurIPS 2025).

Multi-Space Texture Refinement pipeline, consecutively refining textures in the latent and the pixel spaces. Extensive experimental results demonstrate that GeneMAN could generate high-quality 3D human models from a single image input, outperforming prior state-of-the-art methods. Notably, GeneMAN could reveal much better generalizability in dealing with in-the-wild images, often yielding high-quality 3D human models in natural poses with common items, regardless of the body proportions in the input images.

# 1   Introduction

Creating high-quality 3D human models is crucial in various real applications, including VR/AR, telepresence, digital human interfaces, film, and 3D game production. Traditional methods [5, 26, 65] usually utilize a dense camera array to capture synchronized posed multi-view images for human reconstruction, which typically involves complicated and time-consuming processes. Towards efficient 3D human reconstruction, many approaches [49, 58, 11, 50, 19, 64, 76, 14] delve into the challenge of reconstructing 3D human models from a single image, which, however, remains an ill-posed problem due to the absence of comprehensive 3D human observation data.

To facilitate 3D human reconstruction, previous methods [78, 68, 67, 19, 76, 14] often employ parametric human models, *e.g.*, SMPL [32] or SMPL-X [40] as 3D human geometry prior. Nevertheless, these parametric models fail to capture 3D clothing details, which impedes the accurate reconstruction of 3D human figures, particularly when individuals are adorned in loose garments. Recently, image-to-3D human reconstruction methods [14, 19, 76] integrate pretrained text-to-image diffusion models [46, 47] to incorporate 2D priors into the reconstruction process. Despite advances in generalizability, single-view 3D human reconstruction has not been fully resolved, especially when dealing with in-the-wild images. The primary challenges are shown in Fig. 1: (a) Varying Body Proportions: many portraits are captured with varying body proportions, such as full-body, half-body, and headshots, while existing methods primarily focus on full-body reconstructions; (b) Human with Objects: in everyday photography, it is common to capture people holding objects, standing on items, or wearing various accessories, which could greatly impact the reconstruction quality. (c) Human Reconstruction with Natural Pose and Textures: due to the absence of a broadly applicable human-specific geometry and texture model, existing methods struggle to reconstruct credible geometry and consistent texture from real-world images. Additionally, the scarcity of high-quality human body data exacerbates the difficulty of tackling the problem.

In this paper, we propose **GeneMAN**, a **Gene**ralizable image-to-3D hu**MAN** reconstruction framework for high-fidelity 3D human reconstruction from a single image. To enhance generalizability, we first collect a comprehensive, multi-source training dataset of high-quality, multi-modal human data, including 3D scans, multi-view videos, single images, and our augmented human data. Based on the multi-source human data collection, human-specific prior models, including a text-to-image diffusion model as the 2D prior and a view-conditioned diffusion model as the 3D prior, have been trained, which could provide more generalizable human priors compared to traditional human parametric models. Leveraging the pretrained human prior models, GeneMAN generates 3D human models through two primary stages. **1) 3D Geometry Reconstruction.** To create precise and intricate 3D human geometry, GeneMAN employs an Initialization-&-Sculpting strategy, which initially predicts a coarse human geometry using NeRF [38], followed by a sculpting process for adding geometry details. **2) 3D Texture Generation.** Utilizing the refined human geometry, GeneMAN implements a multi-space texture refinement pipeline to produce high-fidelity, consistent 3D textures. Initially, coarse textures are created through multi-view texturing and iteratively refined in the latent space. Subsequently, detailed 3D textures are achieved via pixel space texture refinement, optimizing the UV maps with a 2D prior-based ControlNet [72]. Extensive experiments demonstrate that GeneMAN surpasses existing state-of-the-art (SoTA) methods, showcasing strong generalization capability and high generation quality. We would like to highlight that GeneMAN could faithfully reconstruct 3D human models with diverse clothing, complex poses, and different personal belongings, given a single in-the-wild image with varying body proportions. Our contributions are summarized as follows:

• We introduce GeneMAN, a generalizable 3D human reconstruction framework built on human-specific prior models trained on collected multi-source data. Our framework enables high-quality 3D human reconstruction regardless of its clothing, pose, or body proportions.

- A few effective 3D human reconstruction modules, such as Geometry Initialization & Sculpting, and Multi-Space Texture Refinement, have been proposed, facilitating template-free 3D human geometry modeling and view-consistency texturing.
- According to experimental results, GeneMAN outperforms previous SoTA methods in single-view 3D human reconstruction, which could also effectively reconstruct high-quality 3D humans for real-world human images.

## 2 Related Work

**Text-to-3D Generation** The rapid advancements in 2D image generation have also greatly accelerated progress in text-to-3D generation. DreamFusion [43] and Magic3D [27] have demonstrated that utilizing pretrained 2D text-to-image diffusion models [48, 46] as guidance can greatly enhance the optimization of 3D representations through Score Distillation Sampling (SDS). Trained on large-scale 2D datasets, these text-to-image models possess excellent capabilities in providing comprehensive 2D knowledge and hallucinating unseen scenes based on specific prompts. Nevertheless, as these models are limited to 2D knowledge, the aforementioned text-to-3D methods are susceptible to multi-view consistency issues. MVDream [55] improves 3D consistency by training a multi-view diffusion model that offers a robust multi-view prior. However, these methods are not well-suited for direct image-to-3D reconstruction tasks because images cannot be accurately captured through textual descriptions, leading to inconsistencies in color and texture across the generated assets.

**Singe Image-to-3D Reconstruction** Single image-to-3D reconstruction has greatly benefited from the thriving advancement of diffusion models. Zero-1-to-3 [30] trains a view-dependent diffusion model on Objaverse [8] by explicitly incorporating the camera parameters into the diffusion process. It enables zero-shot image-conditioned novel view synthesis and facilitates general image-to-3D reconstruction with consistent geometric view. Magic123 [44] and DreamCraft3D [61] integrate the 3D prior from Zero-1-to-3 [30] with 2D prior from text-to-image diffusion model, employing a coarse-to-fine two-stage optimization to achieve high-quality 3D reconstruction. LRM [17] leverages both synthetic data from Objaverse [8] and multi-view data from MVImageNet [71] to train a transformer-based 3D architecture, enhancing its generalizability for 3D reconstruction. However, these general image-to-3D approaches often yield poor results in human reconstruction, resulting in inaccurate geometry of human bodies and the neglect of intricate details, such as facial features and clothing. This limitation arises from the lack of human-specific priors.

**Template-based 3D Human Reconstruction.** Template-based approaches rely on human parametric models [32, 41, 69] to guide 3D reconstruction [78, 68, 67, 20, 12, 74, 76, 14]. PaMIR [78] and ICON [68] leverage SMPL-derived features to facilitate implicit surface regression, while ECON [67] employs explicit representations, predicting normals and depths to generate 2.5D surfaces. GTA [74] utilizes transformers with learnable embeddings to translate image features into 3D triplane features. Several methods [19, 14, 76] exploit the generative capabilities of pretrained diffusion models [46] for 3D human reconstruction. SiTH [14] finetunes an image-conditioned diffusion model to hallucinate back views, while SIFU [76] and TeCH [19] utilize diffusion-based geometry and texture refinement to improve reconstruction quality. Recently, IDOL [79] predicts Gaussian-based representation [22] of 3D human to enable fast reconstruction. However, template-based methods are highly reliant on precise human parametric models, which are frequently unattainable for in-the-wild human images.

**Template-Free 3D Human Reconstruction.** Without parameterized human models, PIFu [49] introduces a novel approach by extracting pixel-aligned features to build neural fields. PIFuHD [50] improves this with high-resolution normal guidance, while PHORHUM [4] jointly predicts 3D geometry, surface albedo, and shading. Recently, HumanSGD [3] leverages pretrained 2D diffusion models [46] as a human appearance prior. HumanLRM [64] trains a diffusion-guided feed-forward model to predict the implicit field of a human without template priors. Nonetheless, they still encounter challenges, such as unrealistic and inconsistent textures, and inferior geometric details, largely due to insufficient priors and overly simplistic model designs.

## 3 Preliminaries

**Diffusion Model.** Diffusion models [15, 57, 59] learn to transform samples from a tractable noise distribution towards a data distribution. They comprise a forward process $\{q_t\}_{t \in [0,1]}$, which pro-

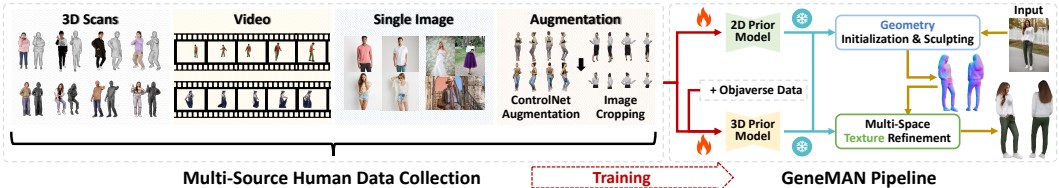

Figure 2: **Overview of the Multi-Source Human Dataset and Our GeneMAN Pipeline.** We have constructed a multi-source human dataset comprising 3D scans, videos, 2D images, and synthetic data. This dataset is utilized to train human-specific 2D and 3D prior models, which provide generalizable geometric and texture priors for our GeneMAN framework. Through geometry initialization, sculpting, and multi-space texture refinement in GeneMAN, we achieve high-fidelity 3D human body reconstruction from single in-the-wild images.

gressively adds random noise to a data point $\boldsymbol{x}_0 \sim q_0(\boldsymbol{x}_0)$, and a reverse process $\{p_t\}_{t \in [0,1]}$, which gradually recovers clean data from noise. The forward process is defined by a conditional Gaussian distribution: $q_t(\boldsymbol{x}_t | \boldsymbol{x}_0) := \mathcal{N}(\alpha_t \boldsymbol{x}_0, \sigma_t^2 \boldsymbol{I})$, where $\alpha_t, \sigma_t > 0$ are time-dependent coefficients. The reverse process is defined by denoising from $p_1(\boldsymbol{x}_1) := \mathcal{N}(\mathbf{0}, \boldsymbol{I})$ with a parameterized noise prediction network $\epsilon_\phi(\boldsymbol{x}_t, t)$ to predict the noise added to a clean data $\boldsymbol{x}_0$, which is trained by minimizing

$$\mathcal{L}_{\text{Diff}}(\phi) := \mathbb{E}_{\boldsymbol{x}_0 \sim q_0(\boldsymbol{x}_0), t \sim \mathcal{U}(0,1), \boldsymbol{\epsilon} \sim \mathcal{N}(\mathbf{0}, \boldsymbol{I})} \\ \left[ \omega(t) \| \epsilon_\phi(\alpha_t \boldsymbol{x}_0 + \sigma_t \boldsymbol{\epsilon}, t) - \boldsymbol{\epsilon} \|_2^2 \right], \tag{1}$$

where $\omega(t)$ is a time-dependent weighting function. After training, we approximate the real data distribution with $p_t \approx q_t$, and thus we can generate samples from $p_0 \approx q_0$.

**Score Distillation Sampling.** Score Distillation Sampling (SDS) is proposed to supervise synthesized novel views by distilling a pretrained text-to-image diffusion model [48] for text-to-3D generation [43]. As the diffusion model is trained on 2D datasets, the SDS loss is referred as $\mathcal{L}_{\text{SDS}}^{2D}$, and its gradient $\nabla_\theta \mathcal{L}_{\text{SDS}}^{2D}$ is formulated as:

$$\nabla_\theta \mathcal{L}_{\text{SDS}}^{2D}(\phi, \boldsymbol{I}) = \mathbb{E}_{t, \boldsymbol{\epsilon}} \left[ \omega(t) (\epsilon_\phi(\boldsymbol{I}_t; y, t) - \boldsymbol{\epsilon}) \frac{\partial \boldsymbol{I}}{\partial \theta} \right], \tag{2}$$

where $\omega(t)$ is a weighting function, $\boldsymbol{I} = g(\theta)$ is the rendered image at random viewpoints, $\boldsymbol{I}_t$ is the noisy image from adding noise $\boldsymbol{\epsilon} \sim \mathcal{N}(0, \boldsymbol{I})$ to the rendered image $\boldsymbol{I}_0$ at timestep $t$, and $y$ denotes the conditioned text prompt. The 3D representation is parameterized by $\theta$, and the diffusion model $\phi$ predicts the sampled noise $\epsilon_\phi(\boldsymbol{I}_t; y, t)$.

A pivotal work, Zero-1-to-3 [30] finetunes a variant of Stable Diffusion [46] model on the Objaverse [8] dataset, resulting in a view-conditioned diffusion model $\phi$, which could synthesize novel view images at any arbitrary viewpoint $\boldsymbol{c}$ given a reference image $\hat{\boldsymbol{I}}$. Hence, $\phi$ offers a strong 3D prior, based on which could derive a 3D-aware SDS loss $\mathcal{L}_{\text{SDS}}^{3D}$. Its gradient $\nabla_\theta \mathcal{L}_{\text{SDS}}^{3D}$ is formulated as:

$$\nabla_\theta \mathcal{L}_{\text{SDS}}^{3D}(\phi, \boldsymbol{I}) = \mathbb{E}_{t, \boldsymbol{\epsilon}} [\omega(t) (\epsilon_\phi(\boldsymbol{I}_t; \hat{\boldsymbol{I}}, \boldsymbol{c}, y, t) - \boldsymbol{\epsilon}) \frac{\partial \boldsymbol{I}}{\partial \theta}]. \tag{3}$$

## 4 Method

In this section, we propose GeneMAN, a generalizable image-to-3D human pipeline aimed for reconstructing high-quality 3D humans from in-the-wild images. By leveraging GeneMAN prior models, which provide human-specific geometric and texture priors, we achieve high-fidelity 3D human reconstruction with varying body proportions, diverse poses, clothing, and personal belongings. The success of GeneMAN prior models is closely tied to the multi-source human dataset we constructed. We first introduce the construction of a multi-source human dataset and GeneMAN prior models in Sec. 4.1 and Sec. 4.2, respectively. Then, we illustrate the details of our GeneMAN framework, including the Geometry Initialization & Sculpting (Sec. 4.3), as well as the Multi-Space Texture Refinement (Sec. 4.4). The overview of GeneMAN framework is shown in Fig. 2.

### 4.1 Multi-Source Human Dataset

The success of LRM [17] has demonstrated that training on multi-source datasets, including the 3D synthetic dataset Objaverse [8] and the video dataset MVImgNet [71], significantly enhances the generalization ability and reconstruction quality of models. However, existing 3D human reconstruction methods typically rely on scarce 3D scanned human data [70, 1] or only video data [77, 66, 7] for training, which restricts their ability to generalize to in-the-wild images, particularly human portraits with diverse poses, clothing, and various body proportions. To address this challenge, we construct a large-scale, multi-source human dataset by collecting diverse human data from 3D scans, multi-view videos, single photos, and synthetically generated human data, as illustrated in Fig. 2.

Our 3D scanned human data is aggregated from the commercial dataset RenderPeople [1], alongside several open-source datasets: CustomHumans [13], HuMMan [6], THuman2.0 [70], THuman3.0 [60] and X-Humans [53]. Additionally, we enrich the dataset by integrating human-specific data filtered from Objaverse [8]. For multi-view human videos, we leverage datasets such as DNA-Rendering [7], ZJU-Mocap [42], AIST++ [24], Neural Actor [29] and Actors-HQ [21]. In terms of 2D human imagery, we select data from DeepFashion [31] and LAION-5B [51] to ensure comprehensive coverage of diverse human appearances. Furthermore, we employ data augmentation strategies to synthesize additional data. Specifically, we utilize ControlNet-based [72] image synthesis to generate multi-view human data with diverse clothing options through various prompts, enabling the creation of human images featuring different garments. To account for varying body proportions, we apply image cropping to preprocessed multi-view human renderings, expanding our dataset to include instances with diverse body proportions. In total, our dataset comprises over $50K$ multi-view instances. Further details of the dataset are provided in the supplementary materials.

### 4.2 GeneMAN Prior Models

Previous studies [27, 61] highlight the complementary benefits of hybrid 2D and 3D priors for 3D reconstruction, where 2D priors provide detailed geometry and texture, and 3D priors ensure multi-view consistency. Inspired by this, we finetune a text-to-image diffusion model [46] and a view-conditioned diffusion model [30] on our multi-source human dataset, serving as GeneMAN 2D and 3D priors for modeling human-specific texture and geometry. For the GeneMAN 3D prior, we finetune the pretrained Zero-1-to-3 [30] model leveraging our collected 3D scans, multi-view videos, synthetic data and 2D images from DeepFashion [31]. Additionally, we incorporate an extra 20% of data curated from the Objaverse [8] dataset. Training on this extensive dataset enables the view-conditioned diffusion model to acquire a generalizable prior for human geometry. Notably, the base Zero-1-to-3 model, originally trained on Objaverse, demonstrates a robust capability for accurately reconstructing objects. This facilitates our finetuned model to handle humans with complex clothing and personal belongings, as discussed earlier. The fine-tuning process is conducted using AdamW [33] optimizer with a learning rate of $10^{-4}$ on eight NVIDIA A100 GPUs for one week. For the GeneMAN 2D prior, we finetune Stable Diffusion V1.5 [46] with our entire multi-source human dataset. An equivalent amount of images from LAION-5B [51] are included to preserve the model's capabilities. Finetuning is performed using AdamW [33] with a learning rate of $10^{-5}$ on four NVIDIA A100 GPUs for five days.

### 4.3 Geometry Initialization & Sculpting

As illustrated in Fig. 3, we adopt hybrid representations, incorporating NeRF [38] and DMTet [54] to reconstruct detailed human geometry from a reference image. Instead of relying on SMPL [32] for initialization, we first train the NeRF network [38] using GeneMAN 2D and 3D prior models to craft a template-free initialized geometry. We leverage Instant-NGP [39] as our NeRF implementation due to its fast inference and ability to recover complex geometry. To supervise the reference view reconstruction, we maximize the similarity between the rendered image and the reference image $\hat{I}$ using the following loss:

$$\mathcal{L}_{\text{ref}} = ||\hat{m} \odot (\hat{I} - g(\theta; \hat{c}))||_2 + ||\hat{m} - g_m(\theta; \hat{c})||_2, \tag{4}$$

where $g(\theta; \hat{c})$ represents the rendered image at the reference viewpoint $\hat{c}$, $\theta$ denotes the NeRF parameters and $\odot$ indicates the Hadamard product. The foreground mask is denoted as $\hat{m}$, and $g_m(\theta; \hat{c})$ renders the silhouette. In addition to supervising the RGB image and mask, we utilize

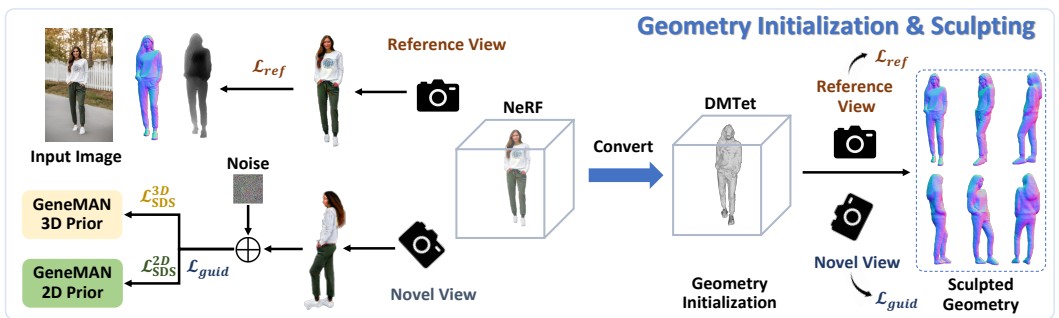

Figure 3: **Geometry Initialization** & **Sculpting.** During the geometry reconstruction stage, we initialize a template-free geometry using NeRF [38], incorporating GeneMAN 2D and 3D priors with SDS losses. Alongside diffusion-based guidance, a reference loss ensures alignment with the input image. We then convert NeRF into DMTet [54] for high-resolution refinement, guided by pretrained human-specific normal- and depth-adapted diffusion models [18].

the geometry prior inferred from the reference image, specifically depth and normal information. The reference depth $\hat{I}_d$ and normal $\hat{I}_n$ are derived using the human foundation model Sapiens [23]. To enforce consistency between the rendered and reference values, we impose depth and normal losses, $\mathcal{L}_{\text{depth}}$ and $\mathcal{L}_{\text{normal}}$. The normal loss is calculated as the mean squared error (MSE) between the reference normal $\hat{I}_n$ and $I_n$, *i.e.*, $\mathcal{L}_{\text{normal}} = ||\hat{I}_n - I_n||_2$. For the depth loss, we employ the normalized negative Pearson correlation:

$$\mathcal{L}_{\text{depth}} = 1 - \frac{\text{cov}(\hat{m} \odot I_d, \hat{m} \odot \hat{I}_d)}{\sigma(\hat{m} \odot I_d)\sigma(\hat{m} \odot \hat{I}_d)}, \tag{5}$$

where $\text{cov}(\cdot)$ and $\sigma(\cdot)$ represent the covariance and variance operators, respectively.

For novel view guidance, we leverage the GeneMAN 2D and 3D prior models, which distill human-specific priors from our multi-source human dataset for hybrid supervision. The GeneMAN 2D prior model $\phi_{2d}$ provides rich details of human geometry and texture, while the GeneMAN 3D prior model $\phi_{3d}$ encodes a pluralistic 3D human prior, ensuring multi-view consistency in both geometry and texture. The hybrid guidance loss $\mathcal{L}_{\text{guid}}$ is denoted as:

$$\mathcal{L}_{\text{guid}} = \mathcal{L}_{\text{2D-SDS}}(\phi_{2d}, g(\theta)) + \mathcal{L}_{\text{3D-SDS}}(\phi_{3d}, g(\theta)). \tag{6}$$

To further refine the geometric details, we adopt DMTet [54], a hybrid SDF-Mesh representation that enables memory-efficient, high-resolution 3D shape reconstruction. The trained NeRF [38] is converted into a mesh, which serves as the geometric initialization for DMTet optimization. During optimization, we compute the reconstruction loss by applying both the MSE loss and the perceptual loss [56] between the rendered normal from the reference view and the reference normal $\hat{I}_n$. Inspired by HumanNorm [18], we leverage its pretrained normal-adapted and depth-adapted diffusion models for novel view guidance to craft high-quality geometry. Furthermore, the SDF loss $\mathcal{L}_{\text{sdf}}$ is incorporated to regularize the geometry, preventing significant deviation from the initial mesh and avoiding improper shapes. Through geometry initialization and sculpting, we achieve high-fidelity 3D human geometry with plausible poses and intricate geometric details.

## 4.4 Multi-Space Texture Refinement

With the detailed geometry in place, we implement multi-space texture refinement in both latent and pixel spaces, ensuring that the textures remain consistent and realistic. First, we perform latent space optimization using the hybrid SDS loss [43, 30], leveraging our GeneMAN 2D and 3D prior models to create an initial coarse texture. The GeneMAN 2D prior model guarantees texture realism, while the GeneMAN 3D prior model excels at maintaining view-consistent textures. We also supervise the reference view by applying an MSE loss between the reference image $\hat{I}$ and the rendered image $g_c(\theta; \hat{c})$ at viewpoint $\hat{c}$. The overall loss function for latent space optimization is formulated as:

$$\mathcal{L}_{\text{coarse}} = \lambda_{ref}^c(||\hat{I} - g_c(\theta; \hat{c}))||_2 + ||\hat{m} - g_c(\theta; \hat{c}))||_2) + \lambda_{guid}^c \mathcal{L}_{\text{guid}}^c, \tag{7}$$

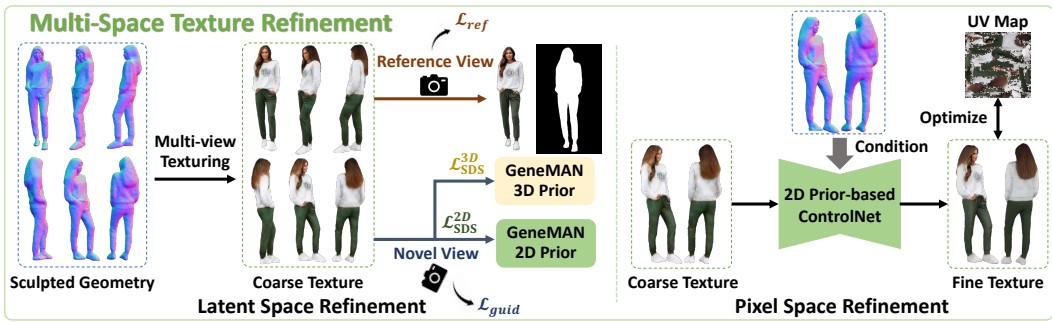

Figure 4: **Multi-Space Texture Refinement.** In the texture generation stage, we propose multi-space texture refinement to optimize texture in both latent space and pixel space. First, we generate the coarse textures using multi-view texturing, which are then iteratively refined in latent space. Subsequently, detailed textures are obtained by optimizing the UV map in pixel space with a 2D prior-based ControlNet.

where $\lambda_{ref}^c$ denotes the weighting parameter of the reference loss, and $\mathcal{L}_{guid}^c$ represents the guidance loss in the coarse texture stage, as defined in Eq. 6.

**Multi-View Training.** Multi-view training strategies effectively enhance consistency across views [30, 55], but they typically require retraining diffusion models. In contrast, we employ a training-free approach by adding identical Gaussian noise to a batch of images rendered from different views. These images are then concatenated into a single image for inference, ensuring that the resulting multi-view images exhibit consistent textures and colors.

The coarse texture stage prioritizes learning consistent and plausible texture but still lacks realism. To enhance texture realism, we employ pixel-space optimization by minimizing the distance between the rendered multi-view images and their refined counterparts, thereby improving the UV texture map. Akin to DreamGaussian [62], we adopt the image-to-image synthesis framework of SDEdit [37] to generate the refined images. Specifically, we render the coarse image $\boldsymbol{I}_{\text{corase}}$ from an arbitrary camera view $c$, perturb it with random Gaussian noise, and apply a multi-step denoising process using the GeneMAN 2D prior based ControlNet [72] $\phi_{2d}(\cdot)$ to obtain the refined image $\boldsymbol{I}_{\text{fine}}$:

$$\boldsymbol{I}_{\text{fine}} = \phi_{2d}(\boldsymbol{I}_{\text{coarse}} + \epsilon(t_{\text{start}}); t_{\text{start}}, e), \tag{8}$$

where $\epsilon(t_{\text{start}})$ represents random noise at timestep $t_{\text{start}}$, and $e$ denotes the conditioned text embeddings. We employ MSE loss for pixel-wise reconstruction and LPIPS [73] loss to enhance texture details. Let $\lambda_{LP}$ denote the weight of the LPIPS loss. The total loss for pixel space optimization $\mathcal{L}_{\text{fine}}$ is:

$$\mathcal{L}_{\text{fine}} = ||\boldsymbol{I}_{\text{fine}} - \boldsymbol{I}_{\text{coarse}}||_2 + \lambda_{LP}\text{LPIPS}(\boldsymbol{I}_{\text{fine}}, \boldsymbol{I}_{\text{coarse}}) \tag{9}$$

## 5 Experiments

### 5.1 Implementation Details

**Training Details.** Our framework is built upon the open-source project ThreeStudio [10]. During the geometry stage, we progressively increase the resolution of NeRF [38] from $256$ to $384$ over $5,000$ steps. We then convert it to an explicit mesh, which serves as the geometry initialization for DMTet [54] at a resolution of $512$. We subsequently optimize DMTet for $3,000$ steps to sculpt fine-grained geometric details. In the texture stage, we perform an initial coarse texture optimization over $10,000$ steps, followed by a refinement of the texture UV map for $1,000$ steps. The full optimization process takes approximately $1.4$ hours on single NVIDIA A100 80G GPU. Additional details are provided in the supplementary materials.

**Testing Details.** For both qualitative and quantitative evaluation, we randomly select $50$ samples from the Internet and CAPE [35]. The samples sourced from the Internet include challenging scenarios, such as humans with varying body proportions (head shots, half-body shots, full-body shots), as well as humans in diverse poses, clothing, and with personal belongings. We compare our model with state-of-the-art image-to-3D human reconstruction methods, including PIFu [49], GTA [75], TeCH [19] and SiTH [14]. For each method, we render $120$ viewpoints across $360$ degrees of the reconstructed results for evaluation. We compute PSNR and LPIPS [73] between the reference view

Table 1: **Quantitative Comparison with State-of-the-art Methods.** Evaluation images are sourced from in-the-wild and the CAPE dataset [35]. Best results are in **bold**, second-best are underlined.

| Method | in-the-wild | | | CAPE | | |
|---|---|---|---|---|---|---|
| | PSNR↑ | LPIPS↓ | CLIP-Sim↑ | PSNR↑ | LPIPS↓ | CLIP-Sim↑ |
| PIFu [49] | 26.968 | 0.035 | 0.594 | 26.912 | 0.028 | 0.764 |
| GTA [74] | 25.060 | 0.064 | 0.568 | **30.376** | 0.019 | 0.785 |
| TeCH [19] | 25.740 | 0.053 | 0.713 | 27.601 | 0.025 | 0.826 |
| SiTH [14] | 20.412 | 0.129 | 0.608 | 21.992 | 0.048 | 0.815 |
| GeneMAN (Ours) | **32.238** | **0.013** | **0.730** | 28.490 | **0.015** | **0.838** |

and the rendered front view to evaluate reconstruction quality. Additionally, CLIP similarity [45] between the reference view and 119 novel views is measured to assess multi-view consistency.

## 5.2 Quantitative Comparison

Tab. 1 presents quantitative metrics comparing GeneMAN with baseline methods on in-the-wild images and the CAPE dataset [35]. For in-the-wild images, GeneMAN outperforms all competing approaches, showcasing its superiority in generalizable human reconstruction, with the highest CLIP-Similarity [45] indicating better multi-view consistency. It also leads in PSNR and LPIPS, highlighting its improved reconstruction quality. On CAPE dataset, GeneMAN continues to excel, achieving the best results in LPIPS and CLIP-Similarity while maintaining competitive performance in PSNR, proving its effectiveness on laboratory dataset as well. Additional geometric results are placed in the supplementary materials.

## 5.3 Qualitative Comparison

We present qualitative results on testing samples from in-the-wild images and the CAPE [35] dataset, showcased in Fig. 6 and Fig. 7, respectively. To better illustrate the reconstruction quality, we visualize the multi-view surface normals and color renderings. PIFu [49] fails to recover realistic geometry and appearance, particularly in side views. Template-based approaches, including GTA [74], TeCH [19] and SiTH [14] heavily rely on precise human pose and shape estimation (HPS). However, HPS methods [9] often produce artifacts such as "bent legs" and struggle with the shapes of children, as seen in both Fig. 6

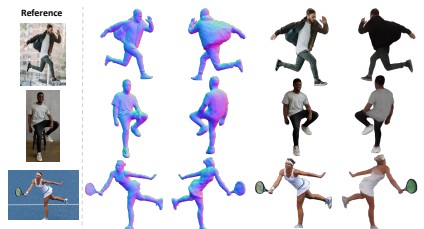

Figure 5: **Qualitative Results of Gene-MANwith Complex Poses.**

and Fig. 7. These limitations lead to cumulative errors in geometry reconstruction that are difficult to eliminate, even with refinement. While TeCH captures more lifelike details with optimization-based refinement, the reconstructed surface appears excessively noisy, and the textures are inconsistent, especially in the back views of garments. In contrast, GeneMAN demonstrates exceptional generalizability to in-the-wild images featuring diverse clothing styles, poses and varying body proportions. Its template-free design enables superior modeling of personal belongings and loose clothing, such as basketball, skirts and dresses, while maintaining detailed, realistic geometry and natural body shapes. GeneMAN also produces high-fidelity, multi-view consistent textures for clothing and hairstyles. In Fig. 5, we present additional results to highlight the robust reconstruction capabilities of GeneMAN under complex poses. More results on complex poses are provided in the supplementary.

## 5.4 User Study

We conduct a user study involving 40 participants to assess the reconstruction quality of GeneMAN compared to each baseline method across 30 test cases, evaluating both geometry and texture quality. Participants are provided with free-view rendering video for each method, and asked to select the most preferred 3D model from five randomly shuffled options. A total of $40 \times 30 = 1200$ comparisons are conducted. As shown in Fig. 10, $73.08\%$ of participants favor our method, demonstrating a significant improvement over the baselines. This indicates that our model generates more plausible and detailed

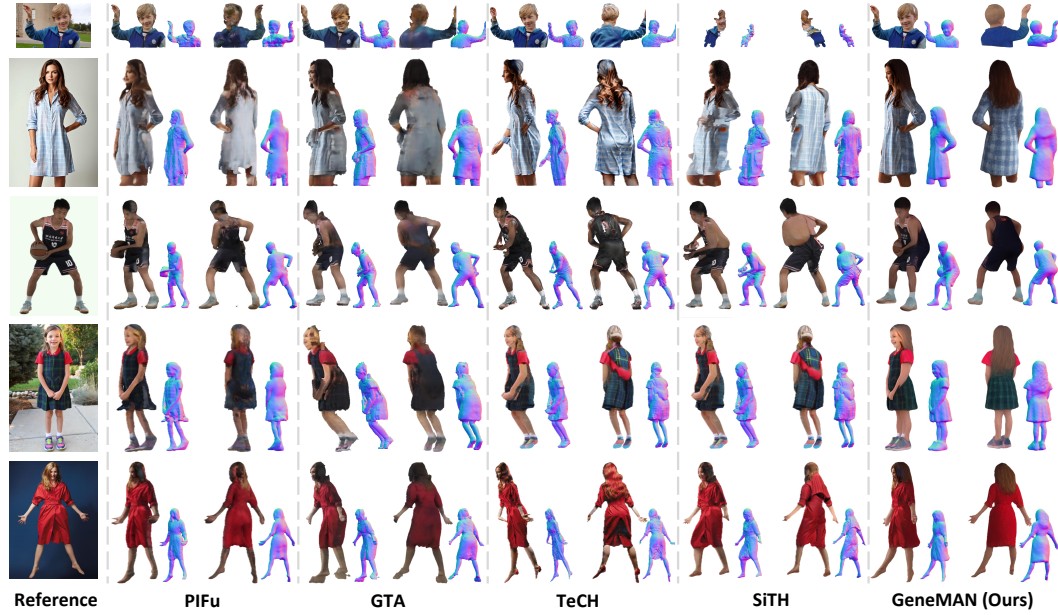

| Reference | PIFu | GTA | TeCH | SiTH | GeneMAN (Ours) |

Figure 6: **Qualitative Comparison with State-of-the-art Methods on in-the-wild Images.** To validate the generalizablity of each method, we select a diverse set of images for demonstration, including human with complex poses, children, varying body proportions and human with personal belongings. GeneMAN shows superiority over compared methods in image-to-3D human reconstruction, achieving both plausible geometry and realistic, consistent texture.

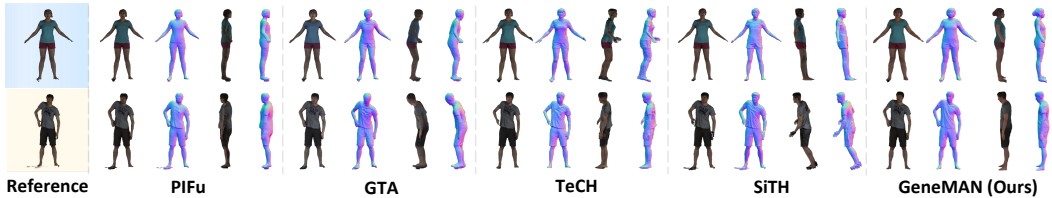

| Reference | PIFu | GTA | TeCH | SiTH | GeneMAN (Ours) |

Figure 7: **Qualitative Comparison with State-of-the-art Methods on CAPE [35].** Without accurate HPS results, template-based methods [74, 14, 19] suffer from artifacts like the "bent-leg" effect. In contrast, GeneMAN reconstructs humans with natural poses, detailed geometry, and realistic textures.

geometry, along with realistic and consistent textures, confirming GeneMAN's effectiveness in generating high-quality reconstructions.

### 5.5 Ablation Study

**Effectiveness of Geometry and Texture Reconstruction.** Fig. 8 analyzes the effectiveness of each stage proposed in our framework. Compared with geometry initialization (a), geometry sculpting in (b) smooths the excessively noisy surface and recovers high-frequency details, such as clothing wrinkles and facial features. In (c), the latent space texturing yields a reasonably satisfactory result; however, the back view remains inconsistent, and the texture appears slightly blurred. To address this, (d) enhance the coarse texture into a final refined texture through pixel space texturing.

**Effectiveness of GeneMAN 2D Prior Model.** To assess the efficacy of GeneMAN 2D prior model, we compare the mesh results extracted from the NeRF [38] stage utilizing the pretrained DeepFloyd-IF [47] model ("original 2D prior") versus GeneMAN 2D prior model ("GeneMAN 2D prior"). Note that we select the NeRF stage output for visualization to clearly demonstrate the impact of the diffusion prior, whereas subsequent geometry and texture disentanglement may obscure this effect. As shown in Fig. 9 (a) and (b), the red-boxed regions reveal that the original 2D prior results in an uneven shirt hem, with the back extending lower than the front. In contrast, GeneMAN 2D prior model ensures multi-view consistency in both geometry and texture, enhancing overall reconstruction.

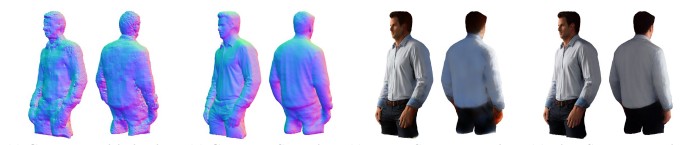

(a) Geometry Initialization    (b) Geometry Sculpting    (c) Latent Space Texturing    (d) Pixel Space Texturing

Figure 8: **Ablation on Geometry and Texture Reconstruction.** (a) shows the geometry initialization, (b) recovers finer details through geometry sculpting, (c) applies latent-space texturing, and (d) achieves the final reconstruction with pixel-space optimization.

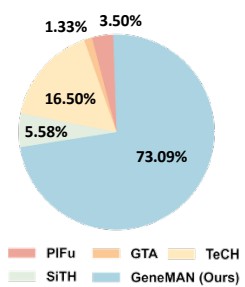

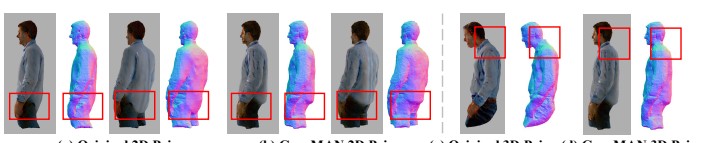

(a) Original 2D Prior    (b) GeneMAN 2D Prior    (c) Original 3D Prior (d) GeneMAN 3D Prior

Figure 9: **Ablation on GeneMAN 2D and 3D Prior Models.** (a) and (b) demonstrate the 2D prior model improves multi-view consistency and texture details. (c) and (d) illustrate the enhanced pose naturalness and geometric accuracy from the 3D prior model.

Figure 10: **User Study.** 73.09% of subjects prefer our method over the baselines in terms of geometry and texture, highlighting the significant superiority of our approach in recovering accurate geometry and high-fidelity appearances.

**Effectiveness of GeneMAN 3D Prior Model.** To assess how GeneMAN 3D prior model ("GeneMAN 3D prior") make influence on reconstruction results, we conduct a comparison by substituting it with the original Zero-1-to-3 [30] model ("original 3D prior") in our framework. As indicated in Fig. 9, (c) reveals unnatural human poses, with the neck and head leaning forward, misaligned with the input image. In contrast, our GeneMAN 3D prior model, finetuned on a large-scale multi-source human dataset, captures a more natural pose and shape prior, as shown in (d). This comparison highlights that our GeneMAN 3D prior offers more reliable and coherent guidance, leading to improved geometric accuracy and a more realistic human structure.

## 6    Conclusion

In this paper, we present GeneMAN, a generalizable framework for single-image 3D human reconstruction. By leveraging the GeneMAN 2D and 3D prior models, trained on a large-scale, multi-source human dataset, GeneMAN is capable of reconstructing high-fidelity 3D human models from in-the-wild images, accommodating varying body proportions, diverse clothing, personal belongings, and complex poses. Extensive experiments validate the effectiveness of our approach, demonstrating its superiority over state-of-the-art methods.

## Acknowledgments

This work was partly supported by National Natural Science Foundation of China (62506229, 62302295), and Natural Science Foundation of Shanghai (25ZR1402268, 2021SHZDZX0102). We sincerely thank Wayne Wu, Shikai Li, and Honglin He for their insightful suggestions and feedback on the manuscript.

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

# A    Implementation Details

## A.1    Camera Sampling

For novel view guidance, we sample the camera distance from the range $\mathcal{U}(3.0, 3.5)$. The elevation angle $\phi$ is drawn from $\mathcal{U}(-10°, 45°)$, and the azimuth angle $\theta$ is uniformly sampled from $[-180°, 180°]$. Additionally, the field of view (FOV) is constrained to the range $[20°, 25°]$, aligning with the fixed camera intrinsics optimized for our finetuned human diffusion models. Building upon HumanNorm [18], we segment the human body into four regions: the head, upper body, lower body, and full body. To enhance part-aware reconstruction with high-fidelity detail, we allocate a sampling probability of $0.7$ to the full body, and $0.1$ to each of the head, upper body, and lower body. We also manually adjust the camera distance for zooming in on specific parts. For instance, when refining the facial region, the camera distance is reduced to $\mathcal{U}(0.8, 1.0)$. For half-body refinement (either upper or lower body), the camera distance is adjusted to $\mathcal{U}(1.5, 2.0)$. Additionally, the camera center is shifted to ensure that the relevant keypoint aligns with the center of the rendered images. To achieve this, we use an off-the-shelf tool to identify 2D keypoints, which are then used during rasterization to back-project the coordinates into 3D space. Note that if a keypoint lies outside the image boundaries, we assign a sampling probability of zero to that point and renormalize the distribution accordingly.

## A.2    Details of Each Stage

**Geometry Initialization** & **Sculpting.**    In the phase of Geometry Initialization, we optimize Instant-NGP [39] from a resolution of $128$ to $384$ over the course of $5,000$ steps. The loss weights for this stage are set as follows: $\lambda_r = 1 \times 10^3, \lambda_m = 100, \lambda_d = 0.05, \lambda_n = 1, \lambda_{2D} = 0.1, \lambda_{3D} = 0.1$. Subsequently, we extract the resulting mesh from Instant-NGP with an isosurface resolution of $256$ as the geometry initialization. During the geometry sculpting stage, we refine the geometry adopting DMTet [54] at a resolution of $512$ for $3,000$ steps, enabling the capture of intricate details of humans. Inspired by HumanNorm [18], we incorporate a progressive positional encoding technique, where the mask on the position encoding for DMTet's SDF features is gradually lifted to introduce higher-frequency components as training progresses. After $2,000$ iterations, the mask fully reveals all positions, allowing the encoding to capture both low- and high-frequency details. Empirically, we observe that omitting progressive positional encoding results in noisy surface reconstructions. The loss weights are set as follows: $\lambda_r = 5 \times 10^3, \lambda_{vgg} = 1 \times 10^3, \lambda_{sdf} = 1.5 \times 10^3$ and $\lambda_{sds} = 1.0$. To ensure consistency between the rendered front view and the input image, we apply a relatively small guidance scale of 20 for novel view generation using the normal- and depth-adapted diffusion models [18]. The noise timestep $t$ is sampled from $\mathcal{U}(0.02, 0.8)$. Besides, we adopt the AdamW [33] optimizer with a base learning rate of $0.01$ during Geometry Initialization and $2 \times 10^{-5}$ during Geometry Sculpting.

**Multi-Space Texture Refinement.**    For latent space texture refinement, we employ SDS optimization [43] in the latent space to optimize the coarse texture at an image resolution of $1024 \times 1024$ for $10,000$ steps. The loss weights for the coarse texture stage are configured as follows: $\lambda_{rgb}^{color} = 1 \times 10^3, \lambda_m^{color} = 100, \lambda_{2D}^{color} = 0.1, \lambda_{3D}^{color} = 0.1$. Following this, we perform pixel-space texture refinement by optimizing the UV texture map for an additional 1,000 steps to achieve finer texture details. To ensure that the added noise does not corrupt the original content while retaining the capacity to enhance image details, we empirically set the starting timestep $t_{start}$ to 0.05 in Eq.8 in the main text. The weight for the VGG loss [56] in the pixel space texture refinement is set to $\lambda_{LP} = 0.01$. We adopt the AdamW [33] optimizer with a base learning rate of $0.01$ in the coarse texture stage and $0.02$ in the fine texture stage.

Table 2: **The Statistics of Multi-Source Human Dataset.** Our multi-source human dataset encompasses a diverse range of human data, categorized into 3D part data: 3D scans, multi-view videos, our synthetic data, and 2D data: single images. The number of 2D images and 3D instances for each dataset is summarized in the table below.

| Multi-Source Human Dataset | | | |
|---|---|---|---|
| **3D Scanned and Synthetic Human Data** | | | |
| RenderPeople [1] | Thuman2.0 [70] | Thuman3.0 [60] | HuMMan [6] |
| 1853 | 526 | 458 | 9072 |
| CostomHuman [13] | X-Humans [53] | Objaverse Human [8] | |
| 647 | 3384 | 3388 | |
| **Video Human Data** | | | |
| DNA Rendering [7] | ZJU-Mocap [42] | Neural Actor [29] | |
| 8780 | 2646 | 4800 | |
| AIST++ [24] | Actors-HQ [21] | | |
| 3853 | 3600 | | |
| **Our Synthetic Human Data** | | | |
| ControlNet-based Synthetic | Image Cropping Synthetic | | |
| 5642 | 3706 | | |
| DeepFashion [31] | LAION-5B [51] | | |
| 20K | 80K | | |

(**3D Part Data** labels the first three sections; **2D Part Data** labels the DeepFashion/LAION-5B section.)

**(a) Three-quarters Body Cropping**          **(b) Half Body Cropping**

Figure 11: **Image Cropping Synthesis.** We apply half-body and three-quarters-length cropping to the multi-view renderings of 3D scans to generate synthetic cropped images.

## B  Details of Multi-Source Human Dataset

To facilitate the training of multi-source human diffusion models, we construct a comprehensive multi-source human dataset containing $100K$ 2D human images and $52,345$ multi-view 3D human instances in total. Detailed statistics of our multi-source human dataset are provided in Tab. 2, with its composition described as follows: (1) The 2D human data consists of $20K$ human images from DeepFashion [31] and $80K$ human images filtered from LAION-5B [51]. The filtering process is conducted using YOLOv7 [63], eliminating images without human subjects or containing multiple individuals. Additionally, images are excluded if they have an aesthetics score below $4.5$, if the largest detected face measures less than $224 \times 224$ pixels, or if the overall resolution falls below $640 \times 1280$. To enhance semantic richness, each image is captioned using a finetuned BLIP model [25]. (2) The 3D human data is collected from 3D scanned human data, synthetic human data, video human data, and our synthetic human data. **3D Scanned Human Data** contains human models sourced from the commercial dataset RenderPeople [1] and open-source datasets including CustomHumans [13], HuMMan [6], THuman2.0 [70], THuman3.0 [60] and X-Humans [53]. **3D Synthetic human data** contains human-category objects filtered from Objaverse [8]. For both scanned and synthetic data, we adopt the dataset creation protocol of Zero-1-to-3 [30], with the exception that we uniformly select 48

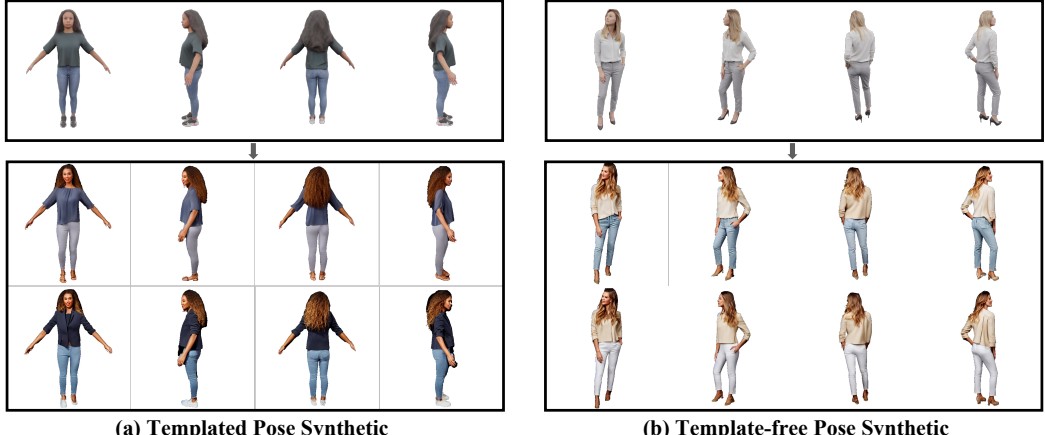

| (a) Templated Pose Synthetic | (b) Template-free Pose Synthetic |

Figure 12: **ControlNet-based Synthesis.** We perform ControlNet-based synthesis for both templated poses and template-free poses.

viewpoints across 360 degrees in azimuth and set the resolution of the rendered image to $1024 \times 1024$. **Video Human Data** contains five open-source datasets: DNA-Rendering [7], ZJU-MoCap [42], Neural Actor [29], AIST++ [24] and Actors-HQ [21]. For datasets containing background imagery, we apply CarveKit [52] to extract human silhouettes and produce corresponding RGBA images. When datasets feature multiple groups of surrounding cameras, we prioritize the group capturing the human subject at the image center. Note that we filter out cases of matting failures in these video datasets. As for DNA-Rendering [7], only Parts 1 and 2 of the released data are utilized. **Our Synthetic Human Data** contains human data synthesized by two augmentation techniques: ControlNet-based [72] image synthesis and image cropping synthesis. To further enhance the diversity of human identities and outfits in the dataset, inspired by En3D [36], we use ControlNet to synthesize a batch of multi-view human data based on diverse prompts describing outfits, genders, ages, and more, all generated by ChatGPT [2]. Unlike En3D, we extract templated or template-free poses and depth images from multi-view renderings of scanned human data (RenderPeople and XHumans [53]) as conditions fed into ControlNet and synthesize multi-view human images using diverse prompts. As shown in Fig. 11 and Fig. 12, examples of Image Cropping Synthesis and ControlNet-based Synthesis are presented, respectively. To ensure the quality of synthetic images, we concatenate pose and depth images horizontally across all views, generating multi-view humans in a single inference. This concatenation strategy ensures the texture consistency of the synthetic multi-view images. To handle the reconstruction of in-the-wild images with arbitrary human proportions, we crop the above fully body-length multi-view 3D human data into half-body length or three-quarters-body length images to create the augmented images. As for 3D data captioning, inspired by 3DTopia [16] and Cap3D [34], we use multi-modal large language model LLaVA [28] to generate captions for 3D objects by aggregating the descriptions from multiple views. The success of our GeneMAN framework demonstrates the effectiveness of our dataset in providing generalized priors for diverse human geometry and textures.

## C    Additional Experiment Results

### C.1    More Qualitative Results

For a more comprehensive geometric evaluation, we incorporated six additional SOTA human geometry reconstruction methods for comparison: PIFuHD [50], PaMIR [78], ICON [68], ECON [67], PHORHUM [4], and SIFU [76]. The geometric comparison results are presented in Fig. 17 and Fig. 18. It can be seen that our model effectively recovers natural human poses, accommodates loose clothing, and excels at reconstructing images with diverse body ratios. In addition, we provide more qualitative comparisons with state-of-the-art methods: PIFu [49], GTA [74], TeCH [19], SiTH [14] as shown in Fig. 19 to Fig. 25. Our method surpasses the compared methods in generating consistent, highly realistic textures with exceptional fidelity.

Besides, in Fig. 13, we conduct additional comparisons with two generalizable methods: HumanLRM [64] and IDOL [79], both of which are feed-forward models trained on large-scale human datasets. As the code of Human-LRM is unavailable, we use the examples provided in its paper. Our model authentically reconstructs 3D humans that closely align with the front view, while IDOL fails to preserve facial details. Our model also produces natural poses with reasonable geometry (see the legs of the first and the third subjects in Fig. 13). Moreover, it achieves high-fidelity appearance with view-consistent details, significantly outperforming HumanLRM.

We provide additional visualization results, with a particular focus on complex poses, in Fig. 15 (in-the-wild images) and Fig. 16 (CAPE [35]). These results highlight the strong generalization ability of GeneMAN to challenging poses.

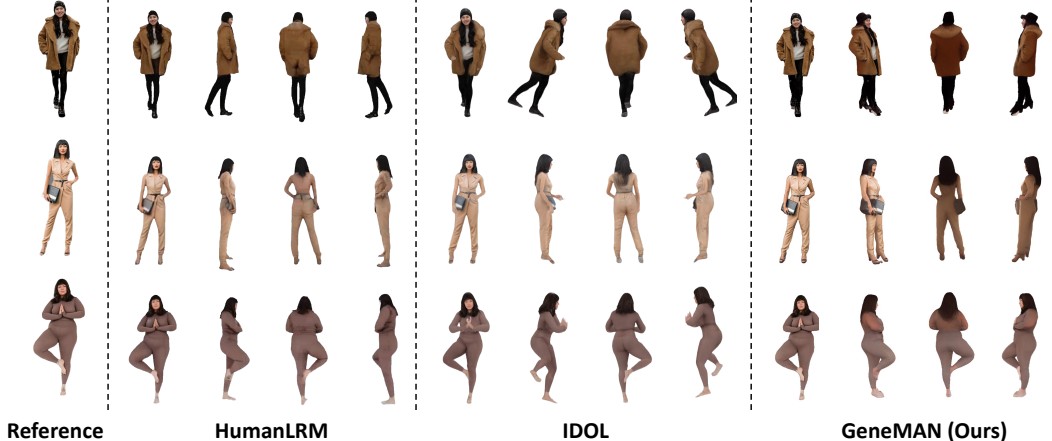

| Reference | HumanLRM | IDOL | GeneMAN (Ours) |

Figure 13: **Additional Comparison on in-the-wild Cases with HumanLRM [64] and IDOL [79].** We conduct experiments on the examples reported by HumanLRM. Best view with zoomed in.

## C.2 Additional Quantitative Comparison Different Across Body Proportions

To assess the robustness of our method under varying human proportions, we additionally evaluate it on in-the-wild images that include head-shot, half-body, and full-body inputs. For comparison, we only consider the template-free method PIFu, since template-based approaches generally fail to reconstruct head-shot and half-body inputs reliably. As shown in Table 3, GeneMAN consistently outperforms PIFu across all proportions setting, demonstrating its strong generalization capability.

Table 3: **Quantitative Comparison Across Different Human Proportions.** The best results are highlighted in **bold**.

| Method | Full-body | | | Half-body | | | Head-shot | | |
|---|---|---|---|---|---|---|---|---|---|
| | PSNR ↑ | LPIPS ↓ | CLIP-Sim ↑ | PSNR ↑ | LPIPS ↓ | CLIP-Sim ↑ | PSNR ↑ | LPIPS ↓ | CLIP-Sim ↑ |
| PIFu | 28.039 | 0.029 | 0.622 | 25.383 | 0.052 | 0.477 | 25.047 | 0.075 | 0.522 |
| GeneMAN (Ours) | **32.529** | **0.012** | **0.751** | **28.245** | **0.016** | **0.681** | **26.374** | **0.022** | **0.661** |

## C.3 Quantitative Geometric Evaluation

Following prior work, we perform a quantitative geometric comparison with baseline methods [49, 78, 68, 67, 74, 19, 14, 76] on the CAPE [35] dataset to assess geometry reconstruction quality. Specifically, we report two commonly used metrics: Chamfer Distance (CD) and Point-to-Surface distance (P2S), both measured in centimeters, between the ground truth scans and the reconstructed meshes. Additionally, to assess the fidelity of reconstructed local details, we calculate the $\mathcal{L}_2$ error between normal images rendered from the reconstructed and GT surfaces, referred to as Normal Consistency (NC). These renderings are obtained by rotating the camera around the surfaces at angles of $\{0°, 120°, 240°\}$ relative to the frontal view.

However, it is crucial to highlight that previous template-based methods [78, 68, 67, 74, 19, 14, 76] directly utilize ground truth SMPL [32, 41] provided in CAPE for evaluation. However, estimating body shape and pose parameters from a single image is an ill-posed problem due to ambiguity, leading to multiple possible solutions, as illustrated in Fig. 14. This presents an unfair advantage over our template-free approach. Moreover, our work focuses on reconstructing 3D humans with high fidelity from in-the-wild images, where accurate SMPL estimates are unavailable. To provide a fair comparison, we re-evaluate the baselines on the CAPE dataset under an inference mode, where *GT body poses are not provided as input*. The lack of access to such ground truth estimations leads to a notable performance drop for template-based methods, which deviates from the results presented in the original papers. The quantitative results are summarized in Tab. 4, where our method outperforms the compared approaches across all geometric metrics, demonstrating the superior reconstruction quality of our method.

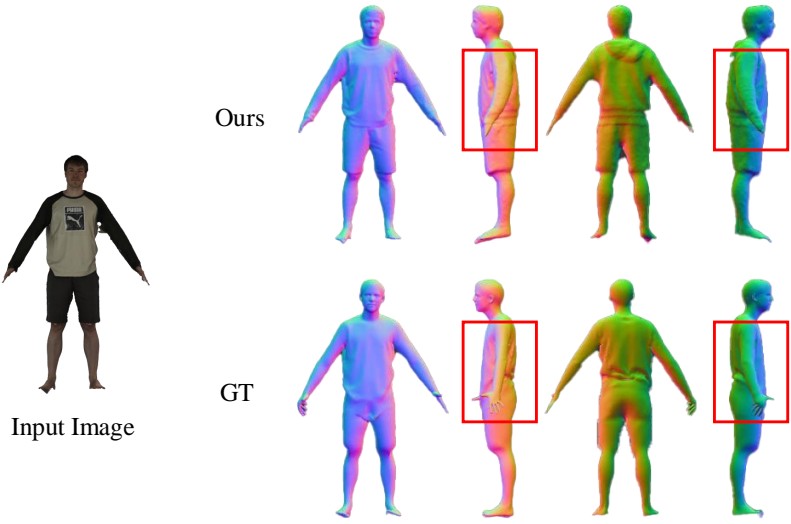

Figure 14: **Visualization Results of the Reconstructed Geometry on the CAPE [35] dataset.** While our template-free method successfully recovers plausible shapes, the orientation of the arms does not perfectly align with the ground truth, as highlighted by the red bounding box. This misalignment is difficult to avoid due to the inherent ambiguity, which can result in large 3D errors. Consequently, it is unfair to compare our approach with SMPL-based methods that utilize GT body parameters directly.

Table 4: **Geometric Comparison with State-of-the-art Methods on the CAPE [35] dataset.** The best results are highlighted in **bold**. The second-place results are underlined.

| Methods | CD ↓ | P2S ↓ | NC ↓ |
|---|---|---|---|
| PIFu [49] | 2.5580 | 2.5770 | 0.0925 |
| PaMIR [78] | 2.5502 | 2.5920 | 0.0925 |
| ICON [68] | 2.4147 | 2.4581 | 0.0872 |
| ECON [67] | 2.0782 | 2.0296 | 0.0798 |
| GTA [74] | 2.6785 | 2.7760 | 0.0914 |
| TeCH [19] | 2.3217 | 2.4163 | 0.0935 |
| SiTH [14] | 1.9182 | 2.0427 | 0.0726 |
| SIFU [76] | 2.5226 | 2.5692 | 0.0875 |
| GeneMAN (Ours) | **1.8862** | **1.9724** | **0.0712** |

## C.4 Ablation of Multi-Source Human Dataset

To validate the effectiveness of our constructed multi-source human dataset, we perform an ablation study on each component: 3D scan human data, video human data, augmented human data, and 2D

human data. Specifically, we compare the reconstruction results using prior models trained with various combinations of these data components. The following experimental settings are evaluated: (a) "Baseline", which replaces 2D and 3D prior models in GeneMAN with their original counterparts (Stable Diffusion V1.5 [46] and Zero-1-to-3 [30]); (b) "Baseline + 3D", which uses prior models trained solely on 3D scanned human data; (c) "Baseline + 3D + Video", which employs prior models trained on both 3D scanned human data and video human data; (d) "Baseline + 3D + Video + AUG" which incorporates prior models trained on 3D scan human data, video human data and augmented human data; (e) "Ours", which utilizes prior models trained on the complete dataset. As detailed in Sec. 5.2 of the main text, we evaluate the performance of each setting using PSNR, LPIPS, and CLIP-Similarity across a set of 30 test cases. As shown in Tab. 5, incorporating 3D scans significantly improves the model's multi-view consistency, resulting in a 0.046 increase in CLIP-similarity. Furthermore, adding video data and augmented data enhances both texture quality and consistency. By utilizing the full dataset, our full-fledged method achieves the best performance in both texture quality and consistency.

Table 5: **The Effectiveness of Multi-source Human Dataset.** The best results are highlighted in **bold**.

| Methods | PSNR ↑ | LPIPS ↓ | CLIP-Sim ↑ |
|---|---|---|---|
| Baseline | 31.503 | 0.018 | 0.662 |
| Baseline + 3D | 30.873 | 0.017 | 0.708 |
| Baseline + 3D + Video | 31.326 | 0.015 | 0.713 |
| Baseline + 3D + Video + AUG | 31.205 | 0.015 | 0.722 |
| GeneMAN (Ours) | **32.238** | **0.013** | **0.730** |

Table 6: **Model Efficiency.** Comparison of inference times between our model and the baselines. Note that PIFu [49], GTA [74], and SiTH [14] are feed-forward methods, whereas TeCH [19] and GeneMAN (ours) are optimization-based approaches that require per-subject optimization. Our GeneMAN model requires a total of 1.42 hours to generate a 3D human asset, with the following time breakdown: Geometry Initialization (22 min), Geometry Sculpting (15 min), Latent Space Texture Refinement (37 min), and Pixel Space Texture Refinement (11 min). All methods are tested on a single NVIDIA A100 80GB GPU.

| | PIFu [49] | GTA [74] | TeCH [19] | SiTH [14] | GeneMAN(Ours) |
|---|---|---|---|---|---|
| **Inference Time** | 4.6s | 24s | 3.5h | 2min | 1.42h |

# D    Limitations

Although our method achieves superior reconstruction performance on in-the-wild images compared to state-of-the-art approaches, it requires a longer optimization time to generate each 3D asset compared to feed-forward methods such as PIFu [49], GTA [74], and SiTH [14], which may limit its practical applicability. Nevertheless, our approach remains faster than TeCH [19], offering a $59.4\%$ increase in efficiency while delivering better quality. The model efficiency of both baseline methods and ours are reported in Tab. 6. Regarding reconstruction quality, our method still struggles to achieve fine-grained modeling for certain parts, such as the hands of full-body humans. Additionally, it lacks specific designs for handling occluded individuals. For human-with-objects reconstruction, our approach primarily focuses on reconstructing personal belongings, but it performs poorly with particularly large or complex objects, such as bicycles. Research on human-with-object reconstruction will be a key focus of our future work.

# E    Broader Impacts and Safeguards

GeneMAN introduces a generalizable framework for reconstructing high-quality 3D human models from a single in-the-wild image. This method is capable of handling various body proportions,

poses, clothing, and personal belongings, offering a wide range of applications, including virtual reality (VR), augmented reality (AR), telepresence, digital human interfaces, film production, and 3D game development. However, the widespread application of GeneMAN may also pose risks. For instance, 3D human reconstruction technology could be misused to generate synthetic content, which may raise ethical and legal concerns. To ensure that the application of GeneMAN is positive and sustainable, we have implemented the following safeguards: 1) Ensure that the technological outcomes of GeneMAN are equitably accessible to diverse social groups, including marginalized communities. 2) Actively collaborate with communities and stakeholders to ensure that technological applications meet societal needs and expectations. 3) Ensure that the development and application of GeneMAN comply with all relevant laws and regulations, including data protection laws and intellectual property rights. 4) Respect all intellectual property rights and ensure that all used data and methods have been appropriately authorized.

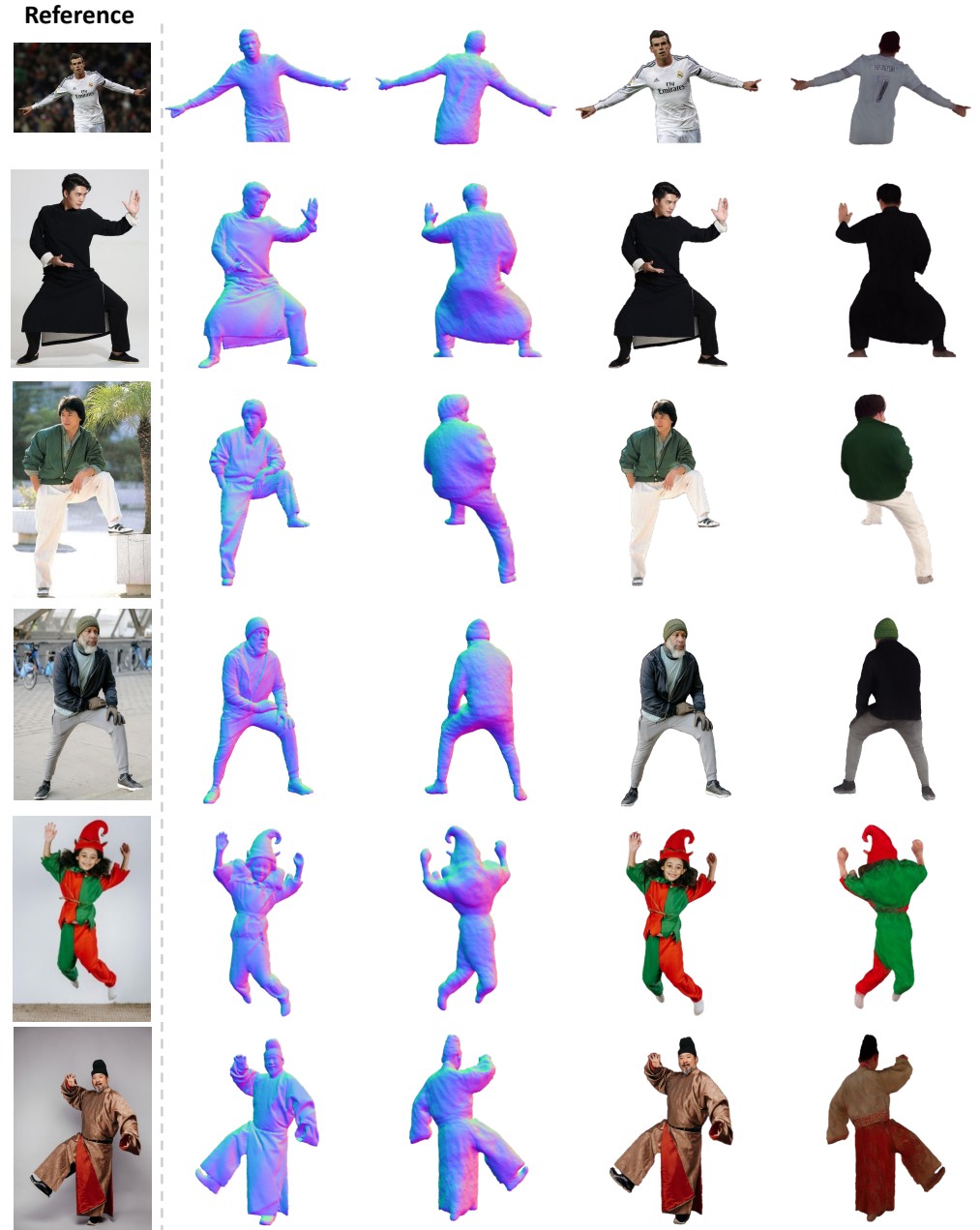

Figure 15: **More GeneMAN Results with Complex Poses on in-the-wild Images.** Best view with zoomed in.

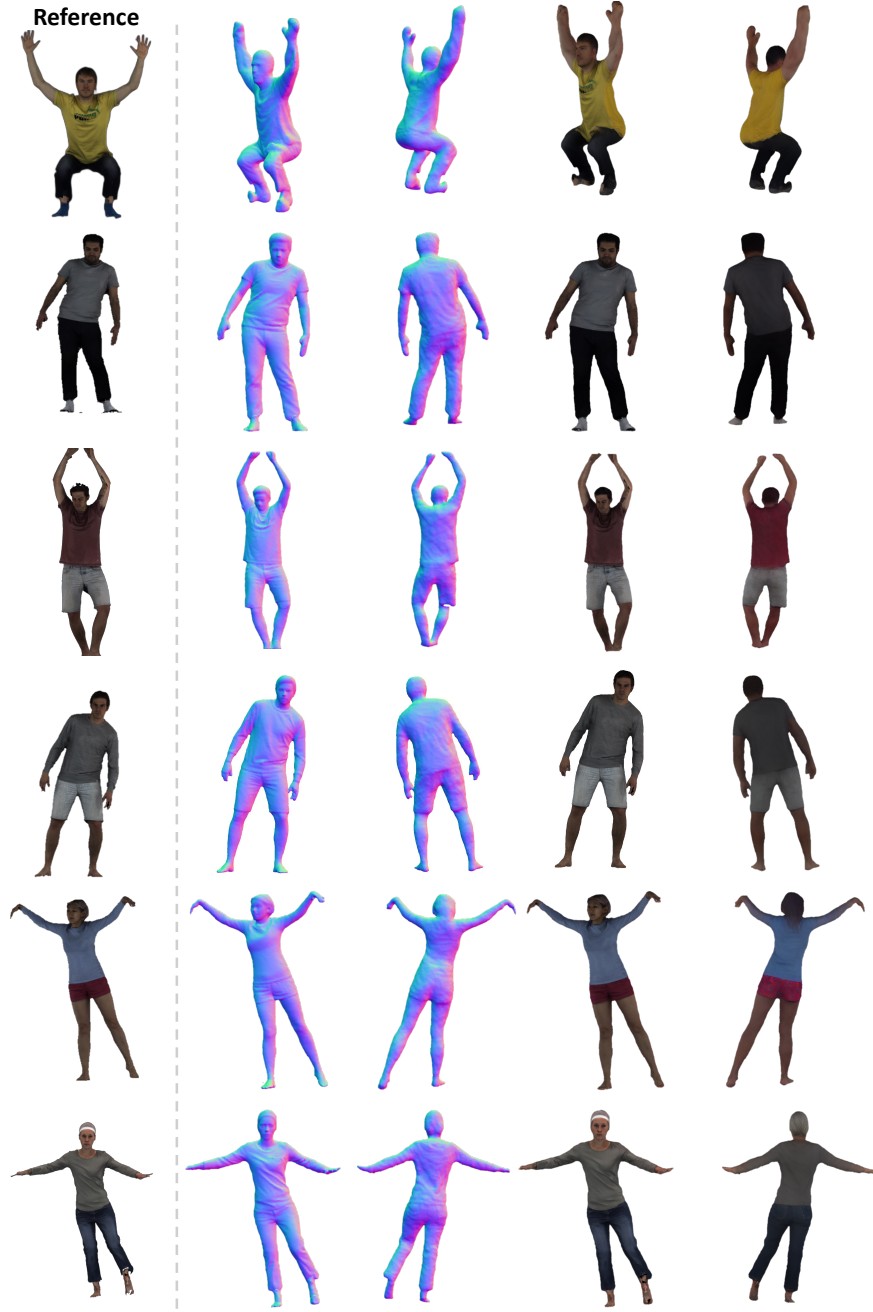

**Reference**

Figure 16: **More GeneMAN Results with Complex Poses on CAPE [35].** Best view with zoomed in.

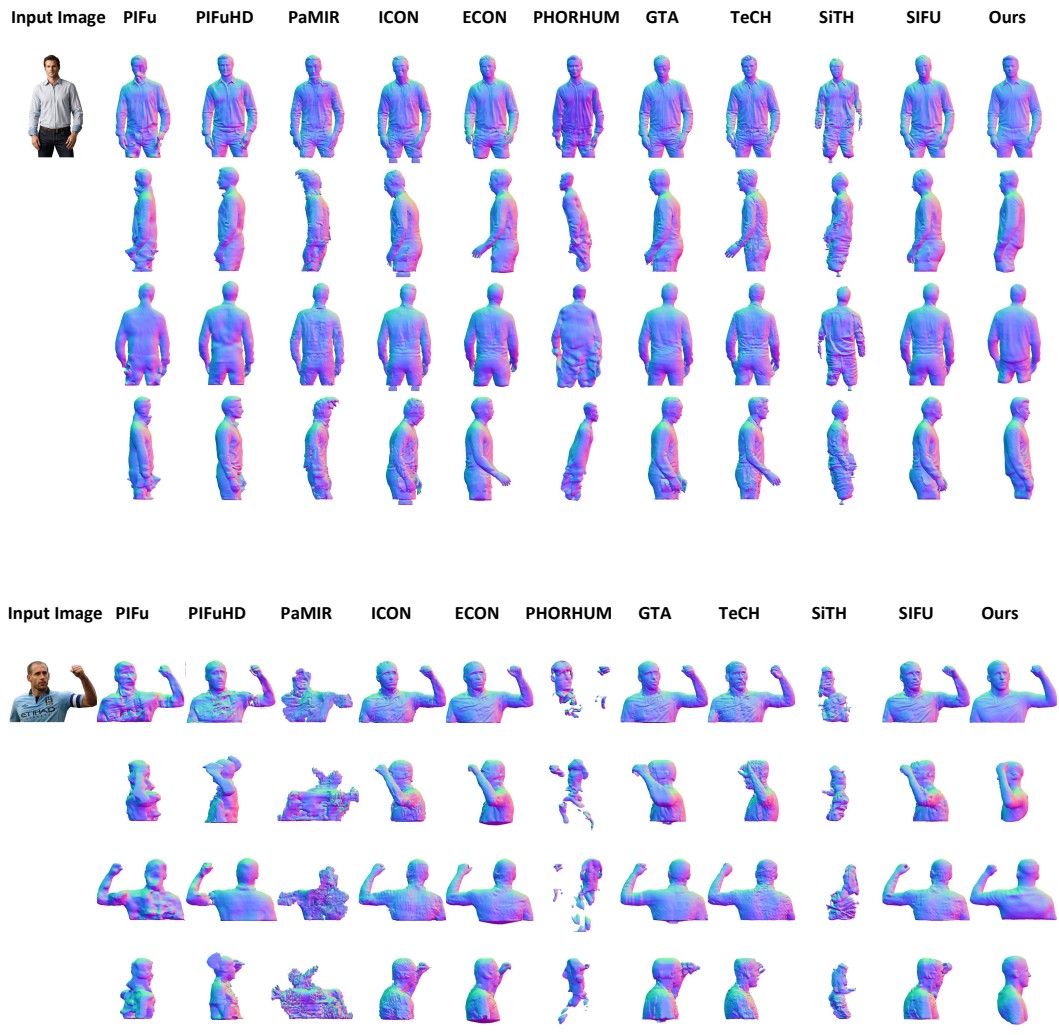

Figure 17: **Geometric Comparison on in-the-wild Images.** Best view with zoomed in.

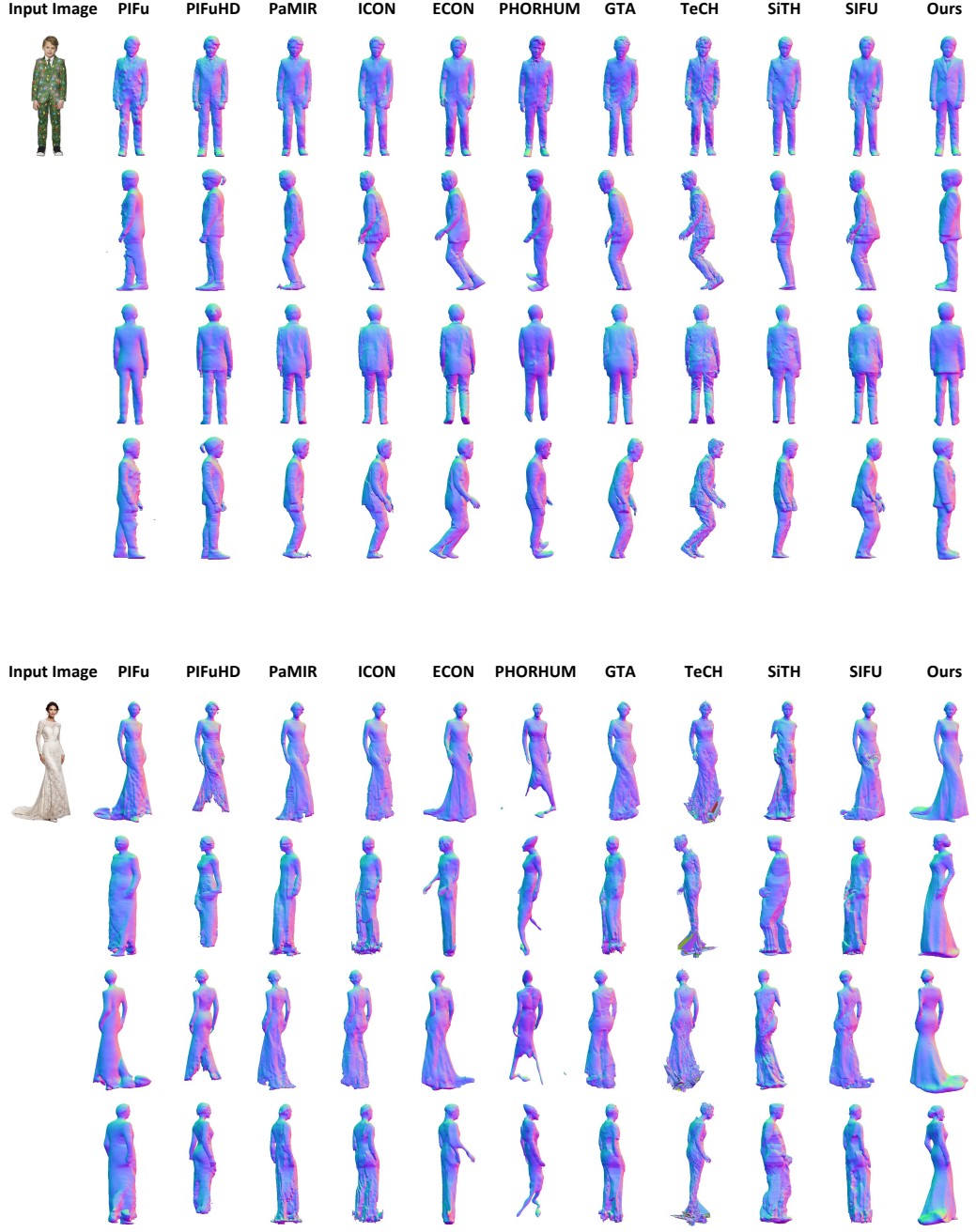

Figure 18: **Geometric Comparison on in-the-wild Images.** Best view with zoomed in.

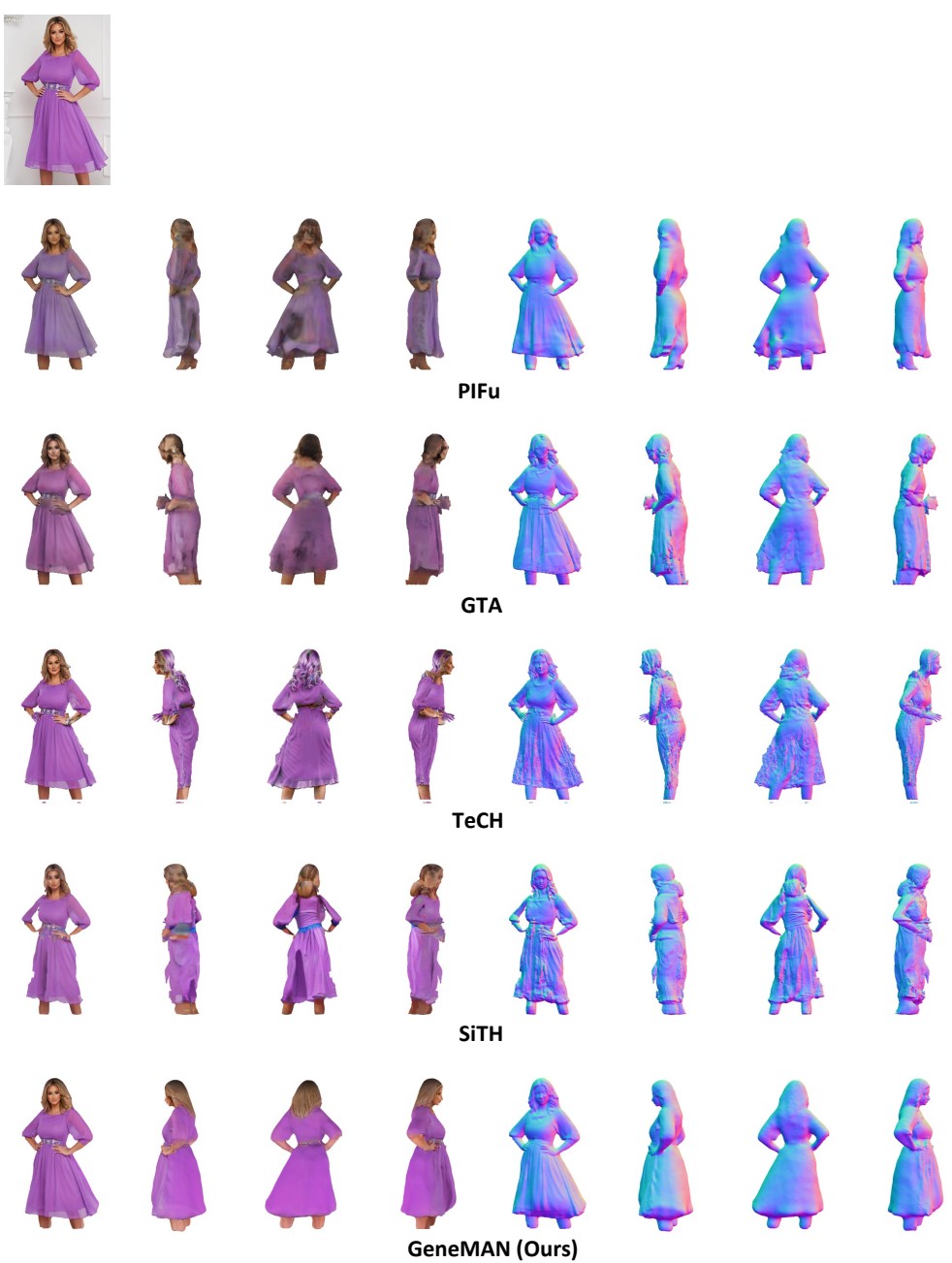

Figure 19: **Qualitative Comparison on in-the-wild Image.** Best view with zoomed in.

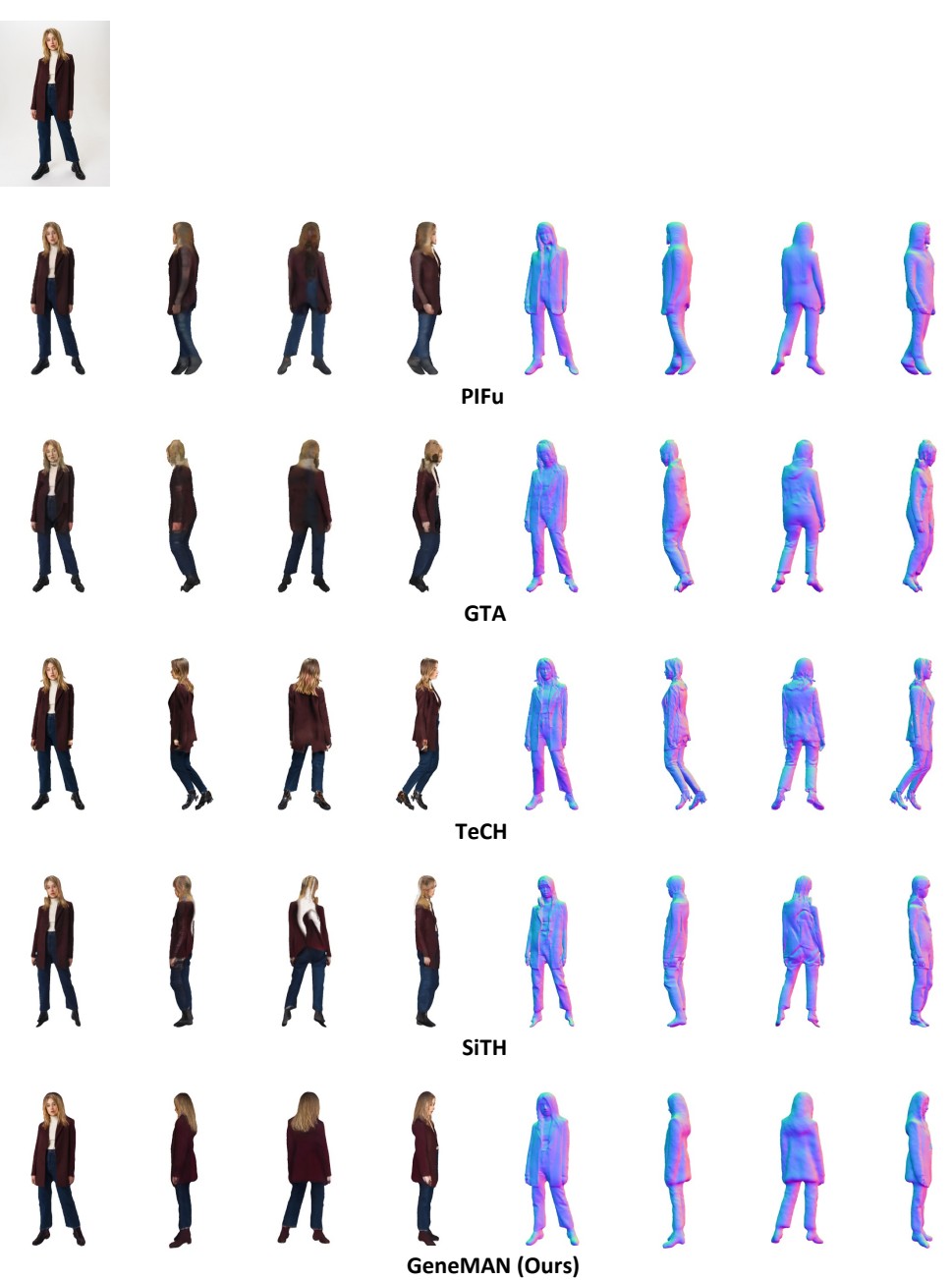

**PIFu**

**GTA**

**TeCH**

**SiTH**

**GeneMAN (Ours)**

Figure 20: **Qualitative Comparison on in-the-wild Image.** Best view with zoomed in.

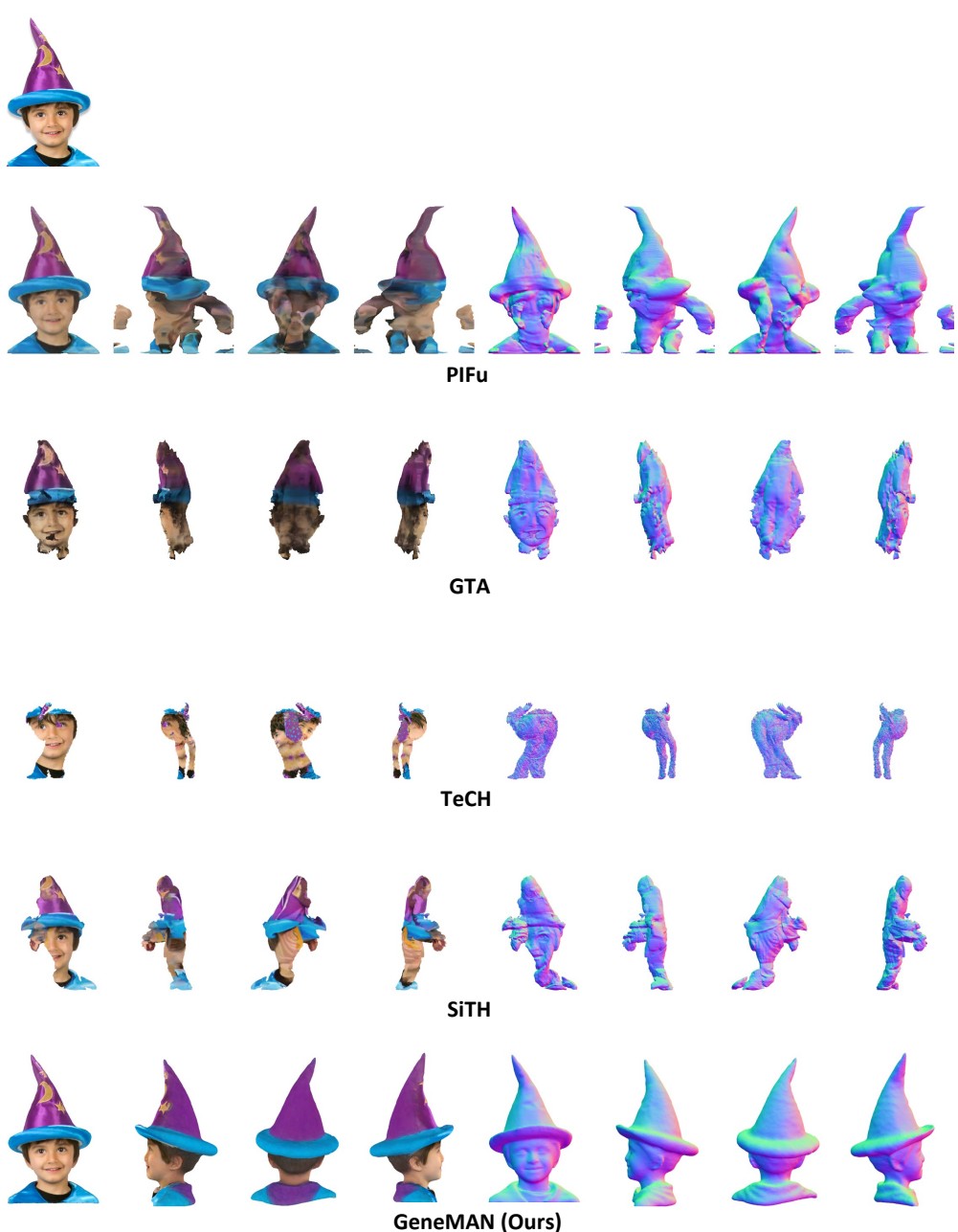

Figure 21: **Qualitative Comparison on in-the-wild Image.** Best view with zoomed in.

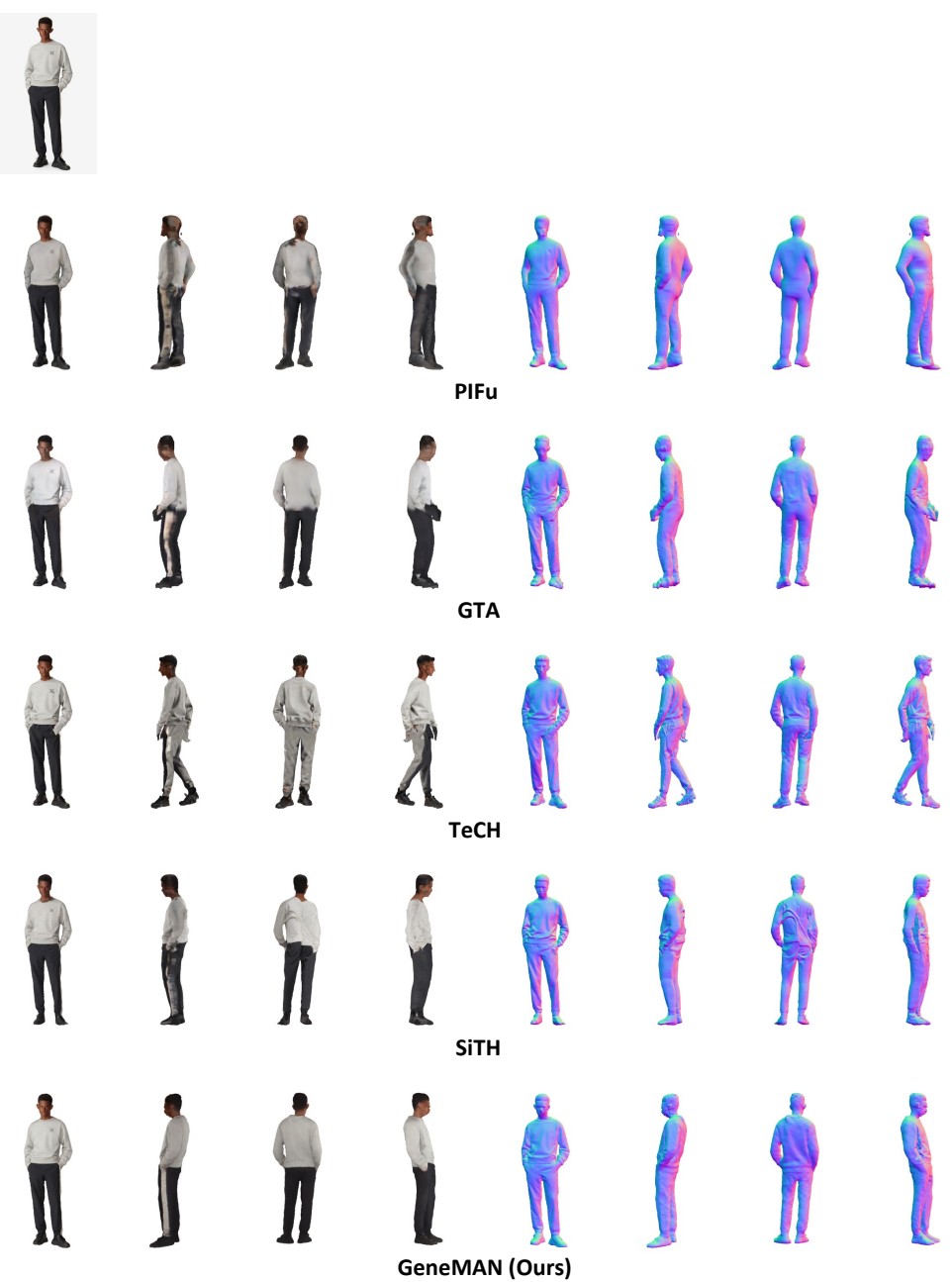

PIFu

GTA

TeCH

SiTH

GeneMAN (Ours)

Figure 22: **Qualitative Comparison on in-the-wild Image.** Best view with zoomed in.

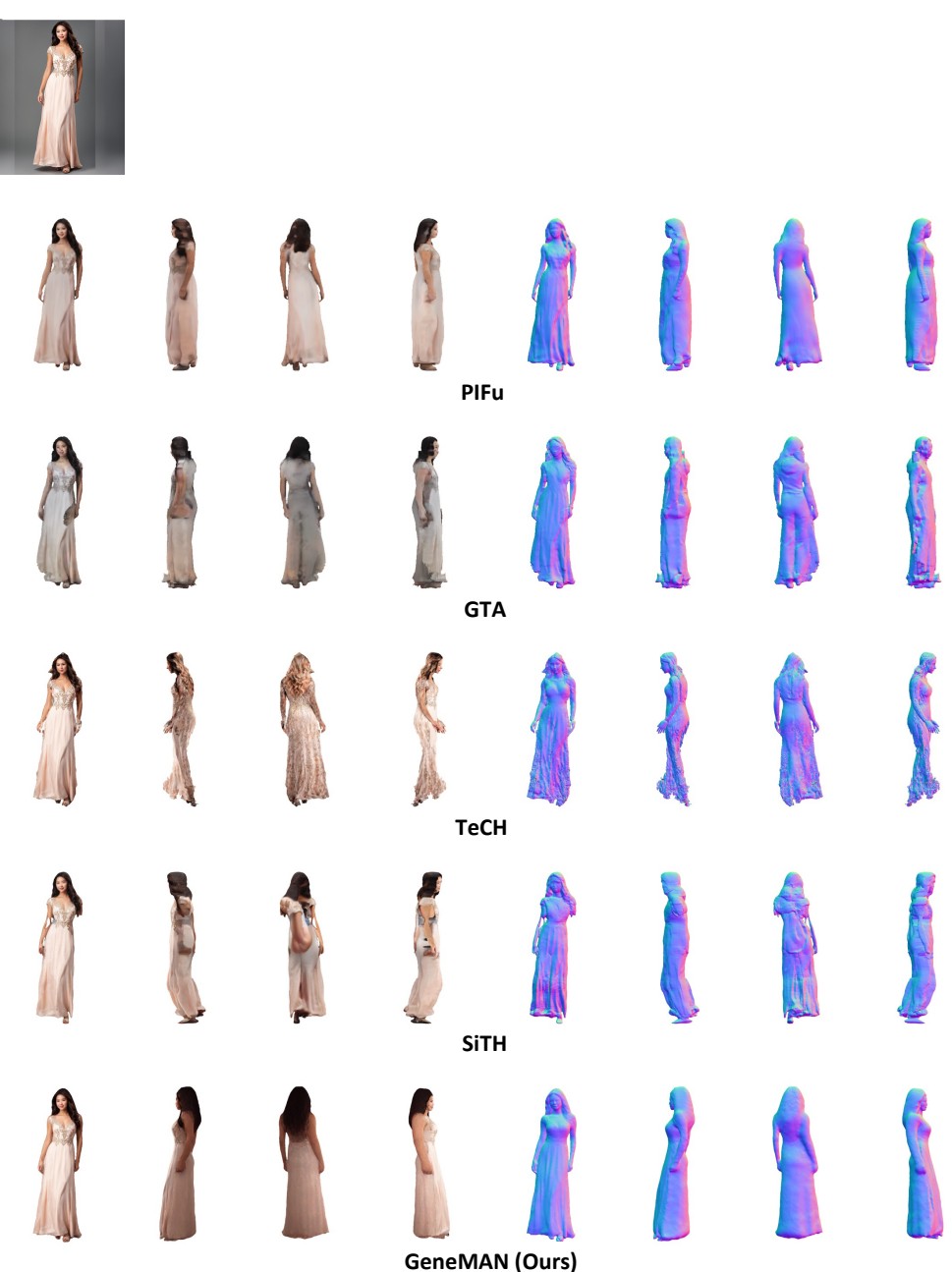

Figure 23: **Qualitative Comparison on in-the-wild Image.** Best view with zoomed in.

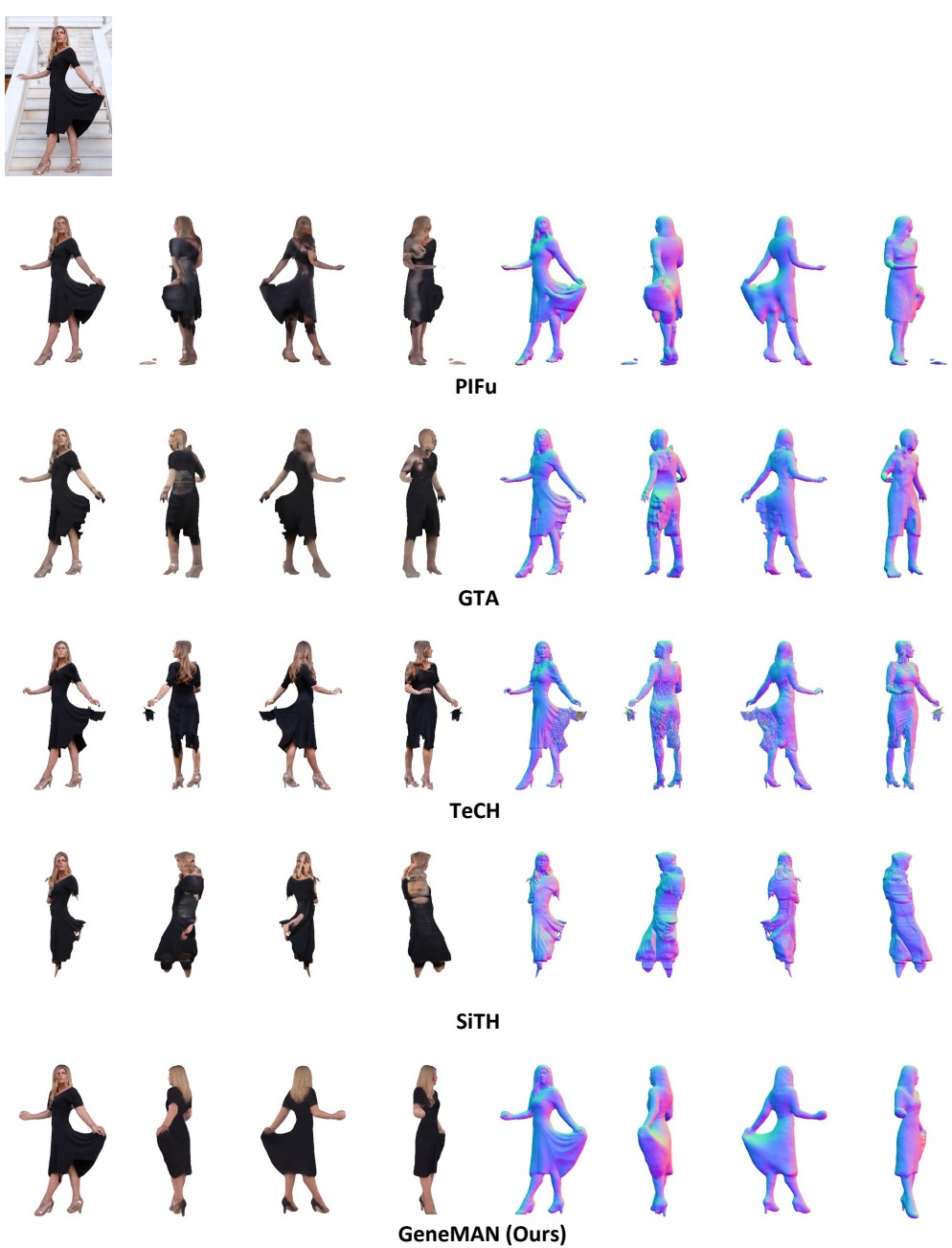

PIFu

GTA

TeCH

SiTH

GeneMAN (Ours)

Figure 24: **Qualitative Comparison on in-the-wild Image.** Best view with zoomed in.

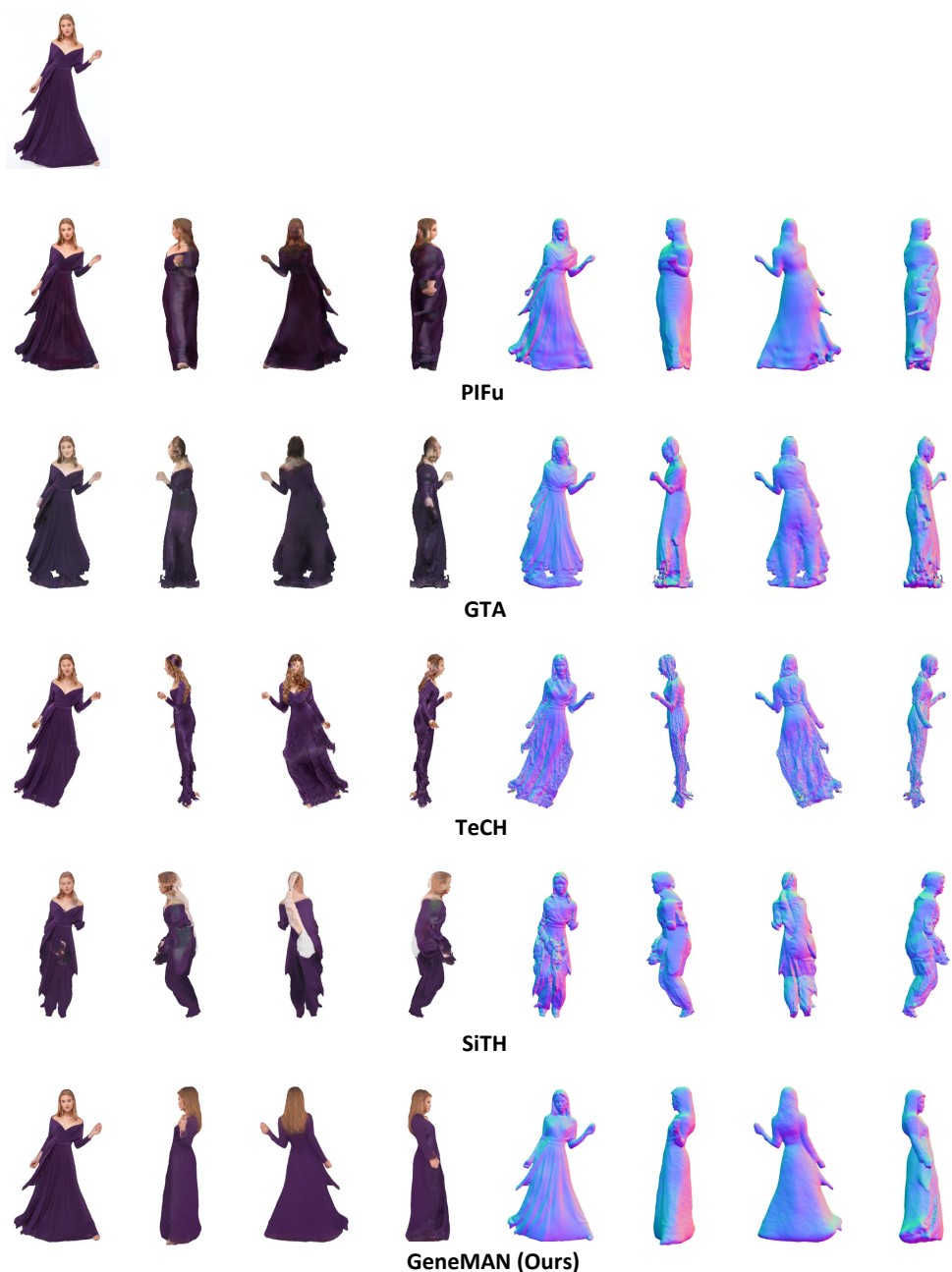

**PIFu**

**GTA**

**TeCH**

**SiTH**

**GeneMAN (Ours)**

Figure 25: **Qualitative Comparison on in-the-wild Image.** Best view with zoomed in.

