# OpenReview forum: "GeneMAN: Generalizable Single-Image 3D Human Reconstruction from Multi-Source Human Data"
_NeurIPS.cc/2025/Conference — NeurIPS 2025 poster_

### Official Review · Reviewer_VcUE · 2025-06-22

**Clarity:** 3
**Significance:** 2
**Originality:** 2
**Rating:** 4
**Confidence:** 3

**Summary:**

This paper introduces a multi-source human dataset aimed at facilitating 3D human reconstruction in the wild. The dataset combines 3D scans, multi-view videos, images, and ControlNet-synthesized data. The authors utilize this dataset to train human-specific 2D and 3D prior models. These priors are used to guide the 3D human reconstruction process through score distillation sampling (SDS) losses. Thanks to the generalizability of the learned priors, the proposed method demonstrates robustness to varying body proportions (e.g., full-body shot, head shot), diverse human belongings, and postures.

**Questions:**

(i) Existing reconstruction methods are typically trained with full-body images, and thus might be biased towards such inputs. Can the authors provide quantitative evaluation results exclusively for head shots, half-body, or full-body inputs? It would be beneficial to understand the advantage of the proposed approach. According to Figure 11 of the supplementary material, it substantially outperforms other SoTA methods on head shots.

(ii) As the proposed method is template-free, it enables joint human-object reconstruction. Figure 1(b) seems to be interesting. How much does the proposed method outperform other template-free methods (e.g., HumanLRM) in those scenarios?

---

I am currently leaning toward `Borderline Reject`, but open to raising the score if the authors clearly show the method's advantages over existing approaches.

**Ethical Concerns:**

["NO or VERY MINOR ethics concerns only"]

**Final Justification:**

The authors have addressed most concerns and included evaluation results demonstrating the proposed method's generalization capability. While HumanLRM [A] appears competitive and faster, it is not yet open-sourced. As the authors will release the GeneMAN model and the multi-source human dataset, this work can provide meaningful value to the research community. I therefore raise my rating to `Borderline Accept`.

- [A] Weng et al., “Template-Free Single-View 3D Human Digitalization with Diffusion-Guided LRM”, arXiv:2401.12175, 2024.

**Limitations:**

yes

**Paper Formatting Concerns:**

There is no major paper formatting concern.

**Quality:**

3

**Strengths And Weaknesses:**

< Strengths >

(i) The multi-source human dataset is comprehensive, capturing a wide variety of human conditions and appearances. This diverse dataset can facilitate in-the-wild 3D human reconstruction. The authors utilize the dataset to build generalizable human-specific prior models, which would benefit the research community.

(ii) The proposed geometry reconstruction method is technically sounding. The SDS losses using the generalizable prior models would be effective to handle diverse scenarios. The authors also utilize depth and normal maps estimated from the human foundation model, Sapiens [A], which is also reasonable.

(iii) The multi-space texture refinement seems to be effective for alleviating over-smoothing or color deviation issues commonly caused by SDS losses. The pixel space refinement is technically sounding.

(iv) The proposed method outperforms existing state-of-the-art reconstruction methods (e.g., TeCH [B], SiTH[C]), as shown in Table 1 and Figures 5-7.

[A] Khirodkar et al., “Sapiens: Foundation for Human Vision Models”, ECCV, 2024.

[B] Huang et al., “TeCH: Text-guided Reconstruction of Lifelike Clothed Humans”, 3DV, 2024.

[C] Ho et al., “SiTH: Single-view Textured Human Reconstruction with Image-Conditioned Diffusion”, CVPR, 2024.

---

< Weaknesses >

(i) While the paper highlights the multi-source human dataset as its main contribution, a significant portion of the data is sourced from existing public datasets, as shown in Table 1 of the supplementary material. Furthermore, existing datasets already offer a broader scale and diversity (e.g., 100K diverse high-fidelity humans in HuGe100K [D]), surpassing the 50K instances in the proposed dataset.

(ii) The proposed reconstruction method appears to be incrementally novel. The geometry initialization is similarly done in Magic123 [E], the geometry sculpting follows a similar approach to HumanNorm [F], and the multi-space texture refinement resembles techniques commonly used in image-to-3D object reconstruction methods. Also, the high-level idea is similar to HumanLRM [G], which also leverages a pre-trained diffusion model for high-quality template-free reconstruction; although it does not use SDS losses.

(iii) As shown in Table 4 of the supplementary material, the proposed method is significantly slower than feed-forward approaches. This raises concerns about its practical applicability. It would be interesting to see the trade-off between reconstruction quality and computational efficiency, particularly in comparison to SoTA feed-forward methods like HumanLRM [G]. As shown in Figure 3 of the supplementary material, HumanLRM appears to achieve competitive reconstruction results at much faster inference speeds.

[D] Zhuang  et al., “IDOL: Instant Photorealistic 3D Human Creation from a Single Image”, CVPR, 2025.

[E] Qian et al., “Magic123: One Image to High-Quality 3D Object Generation Using Both 2D and 3D Diffusion Priors", ICLR, 2024.

[F] Huang et al., “HumanNorm: Learning Normal Diffusion Model for High-quality and Realistic 3D Human Generation”, CVPR, 2024.

[G] Weng et al., “Template-Free Single-View 3D Human Digitalization with Diffusion-Guided LRM”, arXiv:2401.12175, 2024.

---

> ### Author Rebuttal · Authors · 2025-07-31
>
> **Q1: While the paper highlights the multi-source human dataset as its main contribution, a significant portion of the data is sourced from existing public datasets, as shown in Table 1 of the supplementary material. Furthermore, existing datasets already offer a broader scale and diversity (e.g., 100K diverse high-fidelity humans in HuGe100K [D]), surpassing the 50K instances in the proposed dataset.**
>
> **A1:** It is important to note that the HuGe100K dataset proposed in IDOL is synthesized using a video generative model. In contrast, our multi-source dataset primarily consists of real-world data (except the augmented portion), which contributes to more natural geometry and appearance reconstruction with high fidelity. Furthermore, HuGe100K is limited to full-body reconstructions, whereas GeneMAN can robustly handle images with varying body proportions. The comparison with IDOL in Figure 13 clearly demonstrates that our method, trained on a diverse multi-source dataset, yields superior results compared to training on synthetic data alone. Besides, IDOL is a concurrent work to ours, and our dataset was developed around the same time as theirs.
>
> **Q2: The proposed reconstruction method appears to be incrementally novel. The geometry initialization is similarly done in Magic123 [E], the geometry sculpting follows a similar approach to HumanNorm [F], and the multi-space texture refinement resembles techniques commonly used in image-to-3D object reconstruction methods. Also, the high-level idea is similar to HumanLRM [G], which also leverages a pre-trained diffusion model for high-quality template-free reconstruction; although it does not use SDS losses.**
>
> **A2:** Thanks for your comments. We leverage certain existing techniques with the intent of enhancing performance, rather than positioning them as core contributions of our approach. Our main contributions span multiple aspects, including the dataset, model design and training strategies, as well as comprehensive experimental results. Specifically:
>
> 1. _**Novel Data Pipeline**_: Our novel data pipeline integrates two data augmentation strategies and is the **first to leverage multi-source human data**—including 3D scans, videos, synthetic datasets, and 2D images—for training. This pioneering use of diverse data sources significantly enhances the effectiveness, scalability, and generalization ability of our method. In contrast, most existing human reconstruction approaches are limited to training on limited amounts of high-quality scans, which restricts both their generalizability and the realism of their results. Even more generalizable methods such as LRM still rely solely on increasing the quantity of scans and synthetic human datasets, without leveraging other modalities like videos or 2D data, thereby limiting their scalability and generalization capacity compared to ours.
>
> 2. _**GeneMAN Prior Models**_: We provide both 2D and 3D prior models trained on our curated multi-source human data, which can be applied to various downstream tasks.
>
> 3. _**Generalization and Results**_: Our method demonstrates strong generalization capabilities and produces consistently high-quality reconstructions, outperforming existing SOTA human reconstruction methods. Moreover, our method shows promising performance on challenging scenarios involving humans with objects.
>
> Finally, we are organizing and preparing our dataset for public release. We also plan to open-source our well-performing models as soon as possible to benefit and contribute to the research community.
>
> **Q3: As shown in Table 4 of the supplementary material, the proposed method is significantly slower than feed-forward approaches. This raises concerns about its practical applicability. It would be interesting to see the trade-off between reconstruction quality and computational efficiency, particularly in comparison to SoTA feed-forward methods like HumanLRM [G]. As shown in Figure 3 of the supplementary material, HumanLRM appears to achieve competitive reconstruction results at much faster inference speeds.**
>
> **A3:** HumanLRM is not open-source, so our comparison is based on results from their official website, which may involve selectively chosen cases. This means we cannot fully evaluate its performance on a broader range of in-the-wild cases. In contrast, GeneMAN is designed to excel in fine-grained geometry and generalization for challenging cases, which can lead to slower inference speeds but offers superior reconstruction quality in complex situations. As shown in Supp. Figure 3, HumanLRM exhibits artifacts, such as unrealistic deformations in the head and beanie, and issues with the backpack and hair, while our method produces more accurate and detailed results. As HumanLRM is not intended to be open-sourced, we will release our method to fill this gap, offering the community a high-quality and generalizable approach for human reconstruction.
>
> **Q4: Existing reconstruction methods are typically trained with full-body images, and thus might be biased towards such inputs. Can the authors provide quantitative evaluation results exclusively for head shots, half-body, or full-body inputs? It would be beneficial to understand the advantage of the proposed approach. According to Figure 11 of the supplementary material, it substantially outperforms other SoTA methods on head shots.**
>
> **A4:** Thank you for the valuable suggestion. We have conducted additional experiments to investigate the performance over in-the-wild images of different body proportions, i.e., head shots, half-body, or full-body inputs. The quantitative metrics are reported in the following tables, showing that our method outperforms PIFu, which is known as one of the most generalizable methods, across all settings. We will include this comparison in the revised version to better highlight our advantages.
>
> **Full-body**
>
> | Method | PSNR   | LPIPS  | CLIP-Similarity |
> |--------|--------|--------|-----------------|
> | PIFu   | 28.039 | 0.029  | 0.622           |
> | Ours   | 32.529 | 0.012  | 0.751           |
>
> **Half-body**
>
> | Method | PSNR   | LPIPS  | CLIP-Similarity |
> |--------|--------|--------|-----------------|
> | PIFu   | 25.383 | 0.052  | 0.477           |
> | Ours   | 28.245 | 0.016  | 0.681           |
>
> **Head-shot**
>
> | Method | PSNR   | LPIPS  | CLIP-Similarity |
> |--------|--------|-----------------|----------------|
> | PIFu   | 25.047 | 0.075 | 0.522           |
> | Ours   | 26.374|0.022|0.661          |
>
>
> **Q5: As the proposed method is template-free, it enables joint human-object reconstruction. Figure 1(b) seems to be interesting. How much does the proposed method outperform other template-free methods (e.g., HumanLRM) in those scenarios?**
>
> **A5:** As noted in A3, HumanLRM is not open-sourced, and despite our attempts to obtain the code by reaching out to the authors, we did not receive a response. Therefore, we can only compare against the results provided in their paper or official website. In addition, we conduct comparisons with other template-free methods, such as PIFu and PIFuHD. It is noteworthy that the majority of recent methods are template-based. Furthermore, we present HOI examples in Figure 1, Figure 5, and Figure 6 in the main text. In the revision, we will include additional HOI examples to further showcase the advantages of GeneMAN in such scenarios.

---

> > ### Comment · Area_Chair_AQfg · 2025-08-05
> >
> > Dear Reviewer VcUE,
> >
> > The authors have responded to the original reviews. Could you read the rebuttal and share your thoughts? Does it address your original concerns? Are there any remaining questions for the authors?
> >
> > Best,
> > AC

---

> > ### Comment · Reviewer_VcUE · 2025-08-05
> >
> > Dear `Authors`,
> >
> > Thank you for the rebuttal. It has addressed most of my comments, and I appreciate the inclusion of evaluation results across different body proportions.
> >
> > My major concern was whether the proposed method offers clear benefits over HumanLRM [A], as HumanLRM appears to produce competitive results while being significantly faster at inference time. However, as the authors pointed out, HumanLRM is not yet open-sourced. At this moment, this work could offer meaningful value to the research community—especially if the authors release the GeneMAN model and the multi-source human dataset.
> >
> > - [A] Weng et al., “Template-Free Single-View 3D Human Digitalization with Diffusion-Guided LRM”, arXiv:2401.12175, 2024.
> >
> > I will raise my score to `Borderline Accept`.
> >
> > Best,
> >
> > `Reviewer VcUE`

---

> > > ### Author Response · Authors · 2025-08-05
> > >
> > > Thank you again for your response and valuable comments. We will release our GeneMAN as soon as possible. If you have any further concerns, please feel free to discuss them with us.

---

### Official Review · Reviewer_itRn · 2025-06-29

**Clarity:** 4
**Significance:** 1
**Originality:** 2
**Rating:** 4
**Confidence:** 5

**Summary:**

In this paper, the authors introduce GeneMAN, a method to reconstruct a 3D model of a human given a single image. The method consists of 3D geometry initialization and 3D texture generation steps, with the former step focusing on constructing a geometrically sculpted NeRF representation and the latter on optimizing and refining the texture of the resulting human. A 3D scanned human dataset is used to train 3D and 2D priors to assist the method, which performs decently well quantitatively and qualitatively. The authors also perform a user study to show that their work is preferred amongst their baselines.

**Questions:**

1. The approach is based on NeRFs instead of Gaussian splatting. While a single GS method is mentioned in the introduction, the reason for selecting NeRFs for this method is unclear. Additionally, the reported computation time of 1.4 hours on an A100 seems to impact the method's practicality —could the authors touch on this point in their introduction and related work?
2. The authors state that templated approaches require precise human parametric models that are “frequently unattainable for in-the-wild human images”. However, the dataset examples do not appear particularly difficult for SMPL fitting, and this claim overlooks a notable advantage of templated methods—their ability to be transformed and animated to new poses. Would the authors consider revisiting or refining this argument to better reflect the strengths and limitations of both approaches?
3. The authors claim that the method operates given an in-the-wild image. Can these images include object occlusions, and how does the method perform when faced with objects blocking the human body? In addition, does this "in-the-wild" term include images from side, back, and top-down views?
4. ControlNet is used to augment data in MoVid. Wouldn’t this method introduce synthetic appearances, negatively effecting the realism of the priors?
5. The method claims to work on numerous body proportions and poses. I recommend conducting a more extensive study of this to validate this point.

**Ethical Concerns:**

["NO or VERY MINOR ethics concerns only"]

**Final Justification:**

The authors addressed many of my points in the rebuttal. I have also reviewed the other reviews, and can see that many of the other reviewers share my sentiments of the method's limited novelty and quality. I am still not convinced of the method's robustness when it comes to object occlusions, and disagree with the authors' claim in the rebuttal that flat faces are "infrequent", as almost every side profile result in Figure 6 has this issue.

With all of this in mind, however, the authors' rebuttal did clarify most of my other questions and issues. I have decided to raise my rating to a borderline accept.

**Limitations:**

yes

**Quality:**

3

**Strengths And Weaknesses:**

Strengths:
1. The overall approach with geometry construction and generation steps makes sense for creating a good 3D representation of a human from 2D.
2. The figures are very clear and make the method and results very easy to understand.
3. The paper itself is clear and well written.

Weaknesses:

1. The visual results of the method are poor. Facial side profiles tend to look flat (see Figure 6), and texture detail in clothing is smoothed out in unseen angles.
2. The approach has limited novelty. The idea of using 2D and 3D priors for single-view reconstruction with SDS has been used by numerous works in the literature. Ablation studies in Figure 9 do not demonstrate very significant improvement based on these priors, suggesting the majority of the successful reconstruction is a result of the base priors and model.
3. The evaluations are insufficient and the results appear to be visually poorer and more computationally expensive than current work. Notably, no Gaussian splatting-based methods are evaluated against. Comparisons should be conducted against DreamGaussian (Tang, 2023), as well as HGM (Chen, 2025), and LHM (Qiu, 2025).

Due to the weaknesses of this work in comparison to the current literature – especially with regards to significance and originality – I recommend borderline reject.

---

> ### Author Rebuttal · Authors · 2025-07-31
>
> **Q1: The visual results of the method are poor. Facial side profiles tend to look flat (see Figure 6), and texture detail in clothing is smoothed out in unseen angles.**
>
> **A1:**  1. **Flat Side Face:** Thanks for your comment. We acknowledge that, due to the absence of strong priors such as the SMPL parametric model, some reconstructions may exhibit a relatively flat appearance in the facial region. However, such cases are infrequent. Moreover, this limitation could be mitigated by incorporating parametric models into our framework. Despite these imperfections, our method still outperforms state-of-the-art approaches in terms of overall reconstruction quality and generalization capability. (More comparisons are presented in the supplementary, including images and videos.)
> 2. **Smoothed Detail:** Thanks for your advice. The details of the normal map can be adjusted via hyper-parameters, allowing for a trade-off between preserving more fine-grained details and achieving smoother surfaces.  In our current experimental setup, we have adopted a relatively strong smoothness regularization as the default setting. In the revised version, we will include additional results showcasing different levels of detail versus smoothing.
>
>
> **Q2: The approach has limited novelty.**
>
> **A2:** **Novelty:** Thanks for your comments. The use of 2D and 3D priors in our method is primarily intended to enhance performance, rather than being positioned as a core contribution of our approach. Our main contributions span multiple aspects, including the dataset, model design and training strategies, as well as comprehensive experimental results. The detailed explanation can be found in our reply to Reviewer JD8y’s Q1.
>
> **Ablation Studies:** The ablation studies presented in Figure 9 are intended to visually demonstrate the contributions of each stage and the two prior models. To further illustrate the improvements brought by the two prior models, we have included additional quantitative experiments.
>
>
>
> **Q3: The evaluations are insufficient and the results appear to be visually poorer and more computationally expensive than current work. Notably, no Gaussian splatting-based methods are evaluated against. Comparisons should be conducted against DreamGaussian (Tang, 2023), as well as HGM (Chen, 2025), and LHM (Qiu, 2025).**
>
> **A3:**  **Reconstruction Quality:** Current human reconstruction methods are still unable to achieve complete and reliable reconstruction on arbitrary examples, especially in complex in-the-wild scenarios. Despite some imperfections, both the main paper and supplementary clearly demonstrate that our approach outperforms existing state-of-the-art methods across several key aspects emphasized in this work: generalizability, geometric and texture consistency, the ability to handle loose clothing and complex poses, adaptation to arbitrary human body proportions, and geometric rationality with respect to the input image.
>
> **Insufficient Experiments:** Actually, we have made an effort to conduct comprehensive experiments to evaluate our method from multiple perspectives. In both the main paper and supplementary, we present quantitative and qualitative results on two datasets (CAPE and in-the-wild images), covering both geometric accuracy and texture quality. We also include evaluations under complex poses to further assess the robustness of our method. For geometry, we compare our method with 10 existing approaches; for texture, we include comparisons with 6 representative methods.
> In selecting the baselines, we aim to cover a diverse set of paradigms. Specifically, we include traditional reconstruction methods such as PIFu, PIFuHD, PaMIR, ICON, ECON, and PHORHUM; optimization-based SDS methods like TECH and SiFu; feed-forward approaches such as HumanLRM, SiTH, and GTA-Human; as well as Gaussian-based methods like IDOL in the supplementary.
>
> Thank you for your suggestion. We will include additional comparisons with more recent Gaussian-based methods, such as LHM, in our experiments. The pretrained models of HGM have not been released so far.   As shown in following Table, while LHM benefits from fast processing speed, its reconstruction quality is significantly inferior, particularly for side and rear views. Additionally, it struggles with complex garments—skirts, for instance, are often incorrectly reconstructed as legs.
>
> | Metric   | PSNR  | LPIPS | CLIP-SIMI |
> |----------|-------|-------|-----------|
> | LHM      | 24.067| 0.019 | 0.726     |
> | Ours     | 32.529| 0.012 | 0.751     |
>
>
> **Q4: While a single GS method is mentioned in the introduction, the reason for selecting NeRFs for this method is unclear.**
>
> **A4:**  NeRF is used only for initialization; our final representation is a mesh. The Gaussian-based method produces better textures but suffers from inferior geometry quality. Although increasing the number of Gaussian kernels can improve its quality, this comes at the cost of reduced speed. Nevertheless, even with such improvements, the geometric quality of Gaussian methods cannot match that of mesh-based approaches.
>
> Unlike Gaussian-based methods, which lack explicit geometry, the mesh representation supports graphics engines and enables a wider range of downstream applications. One limitation of Gaussian Splatting (GS) is that it heavily depends on the initialization and struggles to adapt to complex topologies. We also experimented with using Gaussian splatting as the primary representation stage but found the results unsatisfactory. In our comparisons with Gaussian-based methods, NeRF-based approaches achieve better geometric accuracy.
>
> **Inference time:**  Although our method is slower than Gaussian-based approaches, it achieves superior geometry and texture quality as well as better generalization capability. Moreover, compared to the state-of-the-art SDS method TECH, which requires 3.5 hours, our method only takes 1.4 hours. Despite the shorter runtime, our approach significantly outperforms TECH across all evaluation metrics. This improvement results from the dedicated design of our network architecture, the integration of multi-source data and training strategies, which form the core contributions of our method and enable both efficient and high-quality reconstruction.
>
> **Q5: The authors state that templated approaches require precise human parametric models that are “frequently unattainable for in-the-wild human images.” However, the dataset examples do not appear particularly difficult for SMPL fitting, and this claim overlooks a notable advantage of templated methods—their ability to be transformed and animated to new poses.**
>
> **A5:**  Thank you for your comment. We agree that SMPL-based methods can produce reasonable estimations for standard images, particularly those with tight clothing and neutral poses. However, in more complex real-world scenarios—such as subjects wearing loose garments, children, elderly individuals, upper-body-only images, or headshots—the template-based paradigm often introduces significant geometric bias. These errors can further propagate during subsequent optimization. For example, in our experiments, we observed that LHM often reconstructs skirts as human legs.
>
> To address this problem, our work adopts a generation-centric, template-free approach, focusing on improving generalizability across diverse and challenging cases. This demonstrates the significance of our method in tackling such reconstruction scenarios and also constitutes one of our key contributions.
>
> While we acknowledge that pose-driven animation and transformation are appealing and important objectives, driving reconstructed avatars across poses remains a long-term challenge. This involves not only modeling human articulation but also handling the dynamic behavior of clothing, which remains far from solved. Template-based methods indeed offer one viable path toward animation, but we believe enhancing the generalization ability of static reconstruction is an equally important and under-explored direction in the field of human modeling.
> In particular, we believe that the generalizable static representations produced by our method provide a strong foundation for future work on animation and 4D reconstruction. We will add a discussion on this aspect in the Future Work section of the revised manuscript.
>
> **Q6:Can these images include object occlusions, and how does the method perform when faced with objects blocking the human body? In addition, does this "in-the-wild" term include images from side, back, and top-down views?**
>
> **A6:** Thanks for your comment. We include challenging cases involving humans with objects and side views in Figure 1, Figure 5, Figure 6 of the main paper, and Figure 15 in the supplementary. As for back and top-down views, they generally lack identifiable features of the person, making the reconstructed identity unreliable and less practical for most real-world applications. Therefore, like most existing methods, our approach primarily focuses on front-facing humans, potentially with some degree of rotation. In our work, generalizability refers more to robustness across diverse clothing styles, age groups , complex poses, human-object interactions, and varying body proportions.
>
> **Q7: ControlNet is used to augment data in MoVid. Wouldn’t this method introduce synthetic appearances, negatively affecting the realism of the priors?**
>
> **A7:** Overusing ControlNet-based synthetic data can indeed negatively impact generation quality. In our experiments, we carefully controlled the proportion of synthetic data to ensure it contributes positively to model performance.
>
> **Q8: The method claims to work on numerous body proportions and poses. I recommend conducting a more extensive study of this to validate this point.**
>
> **A8:** Thank you for your suggestion. We have conducted an additional quantitative comparison for the ablation study, shown in the following table.

---

> > ### Comment · Reviewer_itRn · 2025-08-05
> >
> > Thank you for the thoughtful rebuttal. My questions have been adequately addressed.
> >
> > I will raise my rating to a borderline accept. I encourage the authors to continue to iterate upon the overly smooth and flat nature of many of their model results and include more diverse input images, especially ones with object occlusions, in the final version of the paper.

---

> > > ### Author Response · Authors · 2025-08-05
> > >
> > > Thank you again for your response and valuable comments.  We will continue to improve our models and include more diverse results in the revised version. In addition, we will provide further quantitative comparisons for the ablation study and incorporate a discussion on template-based and template-free methods in the future work. If you have any further questions, please feel free to discuss them with us.

---

### Official Review · Reviewer_rceA · 2025-06-29

**Clarity:** 2
**Significance:** 2
**Originality:** 3
**Rating:** 4
**Confidence:** 5

**Summary:**

This paper proposed a generalizable framework for single-image 3D human reconstruction names GeneMAN.

Giving a single-view image, the method first obtains hybrid representations 3D human with NeRF and DMTet by using 2D and 3D prior models. Then their use geometry sculpting and texture refinement to enhance the results.

**Questions:**

The process of Fig. 3 is not clear, and the relationships between different parts are chaotic. For example, the author uses at least 6 different arrows, what does each arrow represent? Especially the various bidirectional arrows are even more confusing.

The comparison seems unfair with GTA, TeCH and SiTH. All three methods apply SMPL(X) as an intermediate variable which may infuse pose-error in final results. The author only shows side-view results in Figure 6. I am wondering the input-view and opposite view comparison results. Furthermore, I hope to see the comparison results with SIFU(CVPR 2024).

In 160-161, the author claims “train on Objaverse improves human geometry results”. The author should use an ablation study to support this point.

**Ethical Concerns:**

["NO or VERY MINOR ethics concerns only"]

**Final Justification:**

The authors’ response has addressed most of my concerns. I will raise my score to borderline accept.

**Limitations:**

yes

**Quality:**

2

**Strengths And Weaknesses:**

Strengths

The process of Geometry Sculpting, and Texture Refinement improve the final results.

From the experimental results, it can be seen that the method proposed by the author outperforms the results of the comparative methods.

Weaknesses

The process of Fig. 3 is not clear, and the relationships between different parts are chaotic. For example, the author uses at least 6 different arrows, what does each arrow represent? Especially the various bidirectional arrows are even more confusing.

The comparison seems unfair with GTA, TeCH and SiTH. All three methods apply SMPL(X) as an intermediate variable which may infuse pose-error in final results. The author only shows side-view results in Figure 6. I am wondering the input-view and opposite view comparison results. Furthermore, I hope to see the comparison results with SIFU(CVPR 2024).

In 160-161, the author claims “train on Objaverse improves human geometry results”. The author should use an ablation study to support this point.

The texture results shown in Figure 5 are not quite consistent. The texture on the input-view to be a direct back projection of the reference image onto the mesh and contains high frequency details, while the texture on the opposite-view to have generated more distinct artifacts.

As the geometry sculpting and texture refinement directly affect by the 3D prior models.  How to ensure viewpoint consistency of 3D prior models ?

In my opinion, the section of 4.1 Multi-Source Human Dataset seems more suitable as part of the related work instead of methods.

From the essence of the model (diffusion model of images), the 2D and 3D prior models proposed by the author seem to have little difference.

Missing references

DIFu: Depth-Guided Implicit Function for Clothed Human Reconstruction, CVPR 2023.

High-fidelity 3D Human Digitization from Single 2K Resolution Images, CVPR 2023.

MVHumanNet: A Large-scale Dataset of Multi-view Daily Dressing Human Captures, CVPR 2024.

PKU-DyMVHumans: A Multi-View Vide

---

> ### Author Rebuttal · Authors · 2025-07-31
>
> We sincerely thank our reviewers for devoting time to this review and offering insightful feedback. The detailed responses are attached as follows.
>
> ### **Weaknesses**
>
> **Q1: The process of Fig. 3 is not clear, and the relationships between different parts are chaotic. For example, the author uses at least 6 different arrows, what does each arrow represent? Especially the various bidirectional arrows are even more confusing.**
>
> **A1:** Thanks for the comment. We have already revised Fig. 3 **in the current version** to improve clarity. Specifically, we have removed all bidirectional arrows. There are now only three distinct types of arrows: the blue arrow indicates the conversion between NeRF and DMTet, while the black arrows represent the steps in the pipeline. Additionally, the two curved black arrows are included solely for typesetting purposes. We hope these changes help to clarify the relationships between the different components of the pipeline. Please refer to the PDF for double checking.
>
> **Q2: The comparison seems unfair with GTA, TeCH and SiTH. All three methods apply SMPL(X) as an intermediate variable which may infuse pose-error in final results. The author only shows side-view results in Figure 6. I am wondering the input-view and opposite view comparison results. Furthermore, I hope to see the comparison results with SIFU(CVPR 2024).**
>
> **A2:** (1) Template-based methods inevitably suffer from errors introduced during SMPL(X) estimation. In contrast, our method does not rely on such templates, offering a more robust approach. Additionally, during our evaluation with in-the-wild images, we ensured that the input to the model remained consistent across all comparisons, maintaining fairness in the evaluation process. (2) Regarding the side view in Fig. 6, we intentionally chose it to highlight the geometry quality. However, to provide a more comprehensive analysis, we have included 360-degree rotating videos in the supplementary materials, which cover the input, side, and opposite views. (3) For a detailed comparison with SIFU, please refer to Tab. 2 and Fig. 7, where we provide both quantitative and qualitative comparisons of geometry performance. It can be seen that our method produces more plausible geometry and finer textures compared to SIFU.
>
> **Q3: In 160-161, the author claims “train on Objaverse improves human geometry results”. The author should use an ablation study to support this point.**
>
> **A3:** We would like to clarify that the statement in lines 160-161 does not suggest that training with 20% Objaverse data improves human geometry reconstruction. The Zero123 model is trained on Objaverse, which provides it with the foundational capability to reconstruct 3D objects. This capability enables our fine-tuned GeneMAN model to handle HOI reconstruction. To ensure training stability and preserve consistency with the data distribution, we fine-tune the Zero123 model by mixing Objaverse data with other sources. However, the primary contribution to human geometry reconstruction comes from training on our curated multi-source human dataset. We apologize for the confusion and will revise our statement in the revision.
>
> **Q4: The texture results shown in Figure 5 are not quite consistent. The texture on the input-view to be a direct back projection of the reference image onto the mesh and contains high frequency details, while the texture on the opposite-view to have generated more distinct artifacts.**
>
> **A4:** The qualitative results in Fig. 5 highlight the advantages of our method over state-of-the-art approaches, particularly when handling complex poses such as running and hopping. GeneMAN demonstrates superior performance in recovering accurate geometry for natural poses, though there is still room for improvement in texture consistency.  Due to the rebuttal format, we cannot include additional visualizations for detailed comparisons, but these will be available in the revised paper. From the results, it is clear that our method consistently avoids the unrealistic limb distortions and noisy surfaces seen in TeCH, SiTH, and GTA. While multi-view texture consistency is inherently a challenging problem in single-image 3D reconstruction, our approach either demonstrates a slight improvement or performs on par with the baselines in this regard.
>
> **Q5: As the geometry sculpting and texture refinement directly affect by the 3D prior models. How to ensure viewpoint consistency of 3D prior models?**
>
> **A5:**  As demonstrated by Zero123, integrating camera parameters into diffusion models provides a scalable solution for achieving view-consistent 3D reconstruction. While there are no explicit 3D representations, the use of implicit controls has proven highly effective when trained on a large amount of multi-view images. Note that our curated dataset includes multi-view human data. Our ablation study in Fig. 9 (c) and (d) demonstrate the model's effectiveness in preserving 3D consistency for reconstructing humans.
>
> **Q6: In my opinion, the section of 4.1 Multi-Source Human Dataset seems more suitable as part of the related work instead of methods.**
>
> **A6:** Thank you for your suggestion. We will move a portion of the "Multi-Source Human Dataset" to the "Related Work" section. Additionally, we will retain a more concise discussion of the dataset within the "Methods" section to better highlight its relevance to our contributions.
>
> **Q7: From the essence of the model (diffusion model of images), the 2D and 3D prior models proposed by the author seem to have little difference.**
>
> **A7:** The architectures of the 2D and 3D prior models differ fundamentally. Specifically, the 3D prior model incorporates camera embeddings as input, enabling it to capture 3D-awareness. For a more in-depth discussion on the necessity and benefits of hybrid priors, you can refer to works such as *Magic123* (Qian et al.) and *DreamCraft3D* (Sun et al.). In brief, 3D priors contribute to enhanced multi-view texture consistency and more accurate geometric representation, while 2D priors, trained on large datasets, excel in generalization and generating high-fidelity details but may lack consistent 3D coherence. Our ablation study, as shown in Fig. 9, further illustrates the complementary effects of both 2D and 3D priors.

---

> > ### Comment · Reviewer_rceA · 2025-08-05
> >
> > I appreciate the  author's efforts and some of my concerns have been addressed.

---

> > > ### Author Response · Authors · 2025-08-05
> > >
> > > Thank you again for your response and valuable comments. We will add the missing reference and polish our writing in the revised version. If you have any further concerns, please feel free to discuss them with us.

---

### Official Review · Reviewer_JD8y · 2025-07-02

**Clarity:** 3
**Significance:** 2
**Originality:** 2
**Rating:** 4
**Confidence:** 3

**Summary:**

This paper focuses on the classic single-view human reconstruction task and proposes a well-designed framework, named GeneMAN, to address the challenges of varying body proportions, diverse personal belongings, and inconsistencies in human texture and posture. The proposed GeneMAN leverages a comprehensive collection of multi-source human data and consists of three main modules: human prior learning using diffusion models, geometry initialization and sculpting, and multi-space texture refinement. Experiments demonstrate that GeneMAN outperforms state-of-the-art methods in generating high-quality 3D human models from single images, exhibiting better generalizability and producing more plausible and detailed geometry with realistic textures.

**Questions:**

Considering the strengths and weaknesses of this paper, I think this is a borderline work. Please refer to the three questions listed in the weaknesses section.

**Ethical Concerns:**

["NO or VERY MINOR ethics concerns only"]

**Final Justification:**

I appreciate the authors for answering my questions in detail in the rebuttal. Considering the contributions and shortcomings of this paper, I am still inclined to give a score of borderline accept. I suggest that the authors carefully revise the paper if it is accepted.

**Limitations:**

Yes.

**Paper Formatting Concerns:**

No.

**Quality:**

3

**Strengths And Weaknesses:**

Strengths:
1. This paper introduces a well-designed method that utilizes various knowledge and skills referred from existing research, including using a comprehensive multi-source collection of high-quality human data, 2D and 3D human priors, and coarse-to-fine strategies for geometry and texture reconstruction, to improve the quality and generalization of single-view 3D human reconstruction.
2. This paper provides sufficient experiments, including quantitative and qualitative results, to demonstrate the effectiveness of the proposed method.
3. This paper is well-organized and easy to read.

Weaknesses:
1. The well-designed method seems somewhat incremental. The novelty of the technical contribution is limited. I think all the modules and training skills can be found in existing work. Is there anything new for this method?
2. Although the proposed method achieves better 3D human reconstruction, it also consumes a longer time (i.e., approximately 1.4 hours on a single A100 GPU) to optimize the results. I think it is a trade-off.
3. Are there any quantitative comparison results for the ablation study? I think it could be better to provide these results.

---

> ### Author Rebuttal · Authors · 2025-07-31
>
> **Q1: The well-designed method seems somewhat incremental. The novelty of the technical contribution is limited. I think all the modules and training skills can be found in existing work. Is there anything new for this method?**
>
> **A1:** Our main contributions lie in multiple aspects, including the curation of a multi-source data pipeline, the design of various training strategies, the promising pretrained 2D and 3D prior models, and strong experimental results. Specifically:
>
> 1. _**Novel Data Pipeline**_: Our novel data pipeline integrates two data augmentation strategies and is the **first to leverage multi-source human data**—including 3D scans, videos, synthetic datasets, and 2D images—for training. This pioneering use of diverse data sources significantly enhances the effectiveness, scalability, and generalization ability of our method. In contrast, most existing human reconstruction approaches are limited to training on limited amounts of high-quality scans, which restricts both their generalizability and the realism of their results. Even more generalizable methods such as LRM still rely solely on increasing the quantity of scans and synthetic human datasets, without leveraging other modalities like videos or 2D data, thereby limiting their scalability and generalization capacity compared to ours.
>
> 2. _**GeneMAN Prior Models**_: We provide both 2D and 3D prior models trained on our curated multi-source human data, which can be applied to various downstream tasks.
>
> 3. _**Generalization and Results**_: Our method demonstrates strong generalization capabilities and produces consistently high-quality reconstructions, outperforming existing SOTA human reconstruction methods. Moreover, our method shows promising performance on challenging scenarios involving human with objects.
>
> Finally, we are organizing and preparing our dataset for public release. We also plan to open-source our well-performing models as soon as possible to benefit and contribute to the research community.
>
> **Q2: Although the proposed method achieves better 3D human reconstruction, it also consumes a longer time (i.e., approximately 1.4 hours on a single A100 GPU) to optimize the results. I think it is a trade-off.**
>
> **A2:** Thank you for your comment. However, we believe this is not simply a trade-off between optimization time and reconstruction quality. A longer optimization time does not necessarily lead to better results. For example, although both our method and TECH are based on SDS optimization, TECH requires 3.5 hours while our method only takes 1.4 hours. Despite the shorter runtime, our method outperforms TECH by a large margin across all evaluation metrics. This improvement results from the dedicated design of our network architecture, the integration of multi-source data and training strategies, which form the core contributions of our method and enable both efficient and high-quality reconstruction.
>
> **Q3: Are there any quantitative comparison results for the ablation study? I think it could be better to provide these results.**
>
> **A3:** Thank you for your suggestion. We have conducted an additional quantitative comparison for the ablation study. As shown in Table 1, using the original 2D/3D prior model results in the lowest performance. Incorporating the 2D prior model effectively improves texture quality, as reflected by metrics such as PSNR and LPIPS. Meanwhile, employing the 3D prior model significantly enhances geometric consistency, measured by CLIP similarity. By utilizing both our 2D and 3D prior models, we achieve the best performance across all metrics.
>
>
> **Table 1: Quantitative comparison results for the ablation study**
>
> | Model            | PSNR   | LPIPS  | CLIP-Similarity |
> |------------------|--------|--------|-----------------|
> | Ori 2D / Ori 3D  | 30.157 | 0.033  | 0.642           |
> | Ours 2D / Ori 3D | 31.856 | 0.018  | 0.696           |
> | Ori 2D / Ours 3D | 31.021 | 0.025  | 0.715           |
> | GeneMAN          | 32.238 | 0.013  | 0.730           |

---

> > ### Comment · Reviewer_JD8y · 2025-08-04
> >
> > Thank you for your detailed response. I have no further questions. I would also like to see the opinions of other reviewers.

---

> > > ### Author Response · Authors · 2025-08-05
> > >
> > > Thank you again for your response and valuable comments. We will add additional quantitative comparison results for the ablation study in the revised version.  If you have any further questions, please feel free to discuss them with us.

---

### Note · Authors · 2025-08-15

**Dear ACs and Reviewers,**

We sincerely thank all reviewers for thoughtful feedback. We appreciate their acknowledgement that our rebuttal effectively addressed their major concerns with additional experiments and clarifications.

---

**Core Contributions**

**Novel Data Pipeline.** We integrate two augmentation strategies and leverage multi-source human data—including 3D scans, videos, synthetic datasets, and 2D images—enhancing effectiveness, scalability, and generalization. Most existing methods rely on limited scans, and even generalizable approaches like HumanLRM omit other modalities, limiting their generalizability.

**GeneMAN Prior Models.** We provide both 2D and 3D prior models trained on our curated multi-source human data, which can be applied to various downstream tasks.

**Generalizability and Quality.** Our method generalizes well across diverse scenarios, outperforms SOTA approaches.

> Note: HumanLRM is not open-sourced. We will release our method to fill this gap for the community.

---

**Main Reviewer Concerns and Responses**

**Novelty.** The novelty of our work lies in multi-source data pipeline, diverse training strategies, pretrained 2D/3D prior models, and strong experimental results (detailed in **Core Contributions**).

**Experimental Sufficiency.**  (a) We performed extensive experiments on CAPE and in-the-wild images, evaluating both geometry and texture, including challenging poses. Our method is compared with a diverse set of baselines covering traditional reconstruction (PIFu, ICON, PaMIR, etc.), optimization-based (TECH, SIFu), feed-forward (HumanLRM, SiTH), and Gaussian-based methods (IDOL). Results demonstrate superior accuracy, quality, and robustness across all settings.
(b) In response to feedback, we add quantitative ablations, body-part evaluations (head, upper, full body), and comparisons with the latest method, LHM.

**Quality and Efficiency.** Our method demonstrates superior quality and generalization across diverse clothing, poses, ages, interactions, and body proportions. Feed-forward methods are fast but lose geometric fidelity and texture consistency (e.g., LHM struggles with complex garments). Compared to SDS-based TECH, our method is faster and more accurate, achieving a trade-off between quality and efficiency.

---

These updates address reviewer concerns and reinforce the novelty, robustness, and relevance of our work. Details are in the rebuttal comments.

Sincerely,

Authors of submission 14057

---

### Decision · Program_Chairs · 2025-09-17

**Decision:**

Accept (poster)

**Comment:**

This paper presents an approach to reconstruct the 3D geometry and appearance of a person from a single image. The method leverages multiple sources of data (e.g., images, videos, and 3D scans) to learn human-based priors and use them during reconstruction. Initially, the paper received one Borderline Accept and three Borderline Reject ratings. Reviewers generally appreciated the quantitative results and the dataset contribution, but expressed concerns that the technical approach was incremental and slow, with questions about its generalization ability and the fairness of the evaluation. The rebuttal initiated discussion between the authors and the reviewers, and it helped clarify several critical concerns. While some issues remain (e.g., limited novelty, uncertainty regarding robustness), the clarifications and additional results provided in the rebuttal addressed the most significant reservations. As a result, the reviewers updated their ratings to four Borderline Accepts. Given the unanimous acceptance recommendation from four knowledgeable reviewers, there is no basis to overturn these reviews. The AC recommends acceptance. Authors should still consider any additional comments or feedback from the reviewers while preparing their final version and of course update the manuscript to include any additional promised changes, analysis, results and/or discussion they provided in their rebuttal.